# Visual Instruction Tuning

**Haotian Liu**[1*]**, Chunyuan Li**[2*]**, Qingyang Wu**[3]**, Yong Jae Lee**[1]
[1]University of Wisconsin–Madison  [2]Microsoft Research  [3]Columbia University
https://llava-vl.github.io

## Abstract

Instruction tuning large language models (LLMs) using machine-generated instruction-following data has been shown to improve zero-shot capabilities on new tasks, but the idea is less explored in the multimodal field. We present the first attempt to use language-only GPT-4 to generate multimodal language-image instruction-following data. By instruction tuning on such generated data, we introduce LLaVA: **L**arge **L**anguage **a**nd **V**ision **A**ssistant, an end-to-end trained large multimodal model that connects a vision encoder and an LLM for general-purpose visual and language understanding. To facilitate future research on visual instruction following, we construct two evaluation benchmarks with diverse and challenging application-oriented tasks. Our experiments show that LLaVA demonstrates impressive multimodal chat abilities, sometimes exhibiting the behaviors of multimodal GPT-4 on unseen images/instructions, and yields a 85.1% relative score compared with GPT-4 on a synthetic multimodal instruction-following dataset. When fine-tuned on Science QA, the synergy of LLaVA and GPT-4 achieves a new state-of-the-art accuracy of 92.53%. We make GPT-4 generated visual instruction tuning data, our model, and code publicly available.

## 1 Introduction

Humans interact with the world through many channels such as vision and language, as each individual channel has a unique advantage in representing and communicating certain concepts, and thus facilitates a better understanding of the world. One of the core aspirations in artificial intelligence is to develop a general-purpose assistant that can effectively follow multi-modal vision-and-language instructions, aligned with human intent to complete various real-world tasks in the wild [4, 26].

To this end, the community has witnessed an emergent interest in developing language-augmented foundation vision models [26, 16], with strong capabilities in open-world visual understanding such as classification [39, 21, 56, 53, 38], detection [28, 61, 32], segmentation [25, 62, 57] and captioning [49, 27], as well as visual generation and editing [41, 42, 55, 15, 43, 29]. We refer readers to the *Computer Vision in the Wild* reading list for a more up-to-date literature compilation [12]. In this line of work, each task is solved independently by one single large vision model, with the task instruction implicitly considered in the model design. Further, language is only utilized to describe the image content. While this allows language to play an important role in mapping visual signals to language semantics—a common channel for human communication, it leads to models that usually have a fixed interface with limited interactivity and adaptability to the user's instructions.

Large language models (LLM), on the other hand, have shown that language can play a wider role: a universal interface for a general-purpose assistant, where various task instructions can be explicitly represented in language and guide the end-to-end trained neural assistant to switch to the task of interest to solve it. For example, the recent success of ChatGPT [34] and GPT-4 [35] have demonstrated the power of aligned LLMs in following human instructions, and have stimulated tremendous interest in developing open-source LLMs. Among them, LLaMA [48] is an open-source LLM that matches the performance of GPT-3. Alpaca [47], Vicuna [9], GPT-4-LLM [37]

37th Conference on Neural Information Processing Systems (NeurIPS 2023).

utilize various machine-generated high-quality instruction-following samples to improve the LLM's alignment ability, reporting impressive performance compared with proprietary LLMs. Importantly, this line of work is *text-only*.

In this paper, we present *visual instruction-tuning*, the first attempt to extend instruction-tuning to the language-image multimodal space, to pave the way towards building a general-purpose visual assistant. In particular, our paper makes the following contributions:

- *Multimodal instruction-following data*. One key challenge is the lack of vision-language instruction-following data. We present a data reformation perspective and pipeline to convert image-text pairs into an appropriate instruction-following format, using ChatGPT/GPT-4.

- *Large multimodal models*. We develop a large multimodal model (LMM), by connecting the open-set visual encoder of CLIP [39] with the language decoder Vicuna [9], and fine-tuning end-to-end on our generated instructional vision-language data. Our empirical study validates the effectiveness of using generated data for LMM instruction-tuning, and suggests practical tips for building a general-purpose instruction-following visual agent. When ensembled with GPT-4, our approach achieves SoTA on the Science QA [33] multimodal reasoning dataset.

- *Multimodal instruction-following benchmark*. We present LLaVA-Bench with two challenging benchmarks, with a diverse selection of paired images, instructions and detailed annotations.

- *Open-source*. We release the following assets to the public: the generated multimodal instruction data, the codebase, the model checkpoints, and a visual chat demo.

## 2   Related Work

**Multimodal Instruction-following Agents.**    In computer vision, existing works that build instruction-following agents can be broadly categorized into two classes: $(i)$ End-to-end trained models, which are separately explored for each specific research topic. For example, the vision-language navigation task [3, 19] and Habitat [46] require the embodied AI agent to follow natural language instructions and take a sequence of actions to complete goals in visual environments. In the image editing domain, given an input image and a written instruction that tells the agent what to do, InstructPix2Pix [6] edits images by following the human instructions. $(ii)$ A system that coordinates various models via LangChain [1] / LLMs [34], such as Visual ChatGPT [52], X-GPT [62], MM-REACT [54], VisProg [18], and ViperGPT [45]. While sharing the same goal in building instruction-following agents, we focus on developing an end-to-end trained language-vision multimodal model for *multiple* tasks.

**Instruction Tuning.** In the natural language processing (NLP) community, to enable LLMs such as GPT-3 [7], T5 [40], PaLM [10], and OPT [59] to follow natural language instructions and complete real-world tasks, researchers have explored methods for LLM instruction-tuning [36, 51, 50], leading to instruction-tuned counterparts such as InstructGPT [36]/ChatGPT [34], FLAN-T5 [11], FLAN-PaLM [11], and OPT-IML [22], respectively. It turns out that this simple approach can effectively improve the zero- and few-shot generalization abilities of LLMs. It is thus natural to borrow the idea from NLP to computer vision. More broadly, the teacher-student distillation ideas with foundation models have been studied in other topics such as image classification [14]. Flamingo [2] can be viewed as the GPT-3 moment in the multimodal domain, due to its strong performance on zero-shot task transfer and in-context-learning. Other LMMs trained on image-text pairs include BLIP-2 [27], FROMAGe [24], and KOSMOS-1 [20]. PaLM-E [13] is an LMM for embodied AI. Based on the recent "best" open-source LLM LLaMA, OpenFlamingo [5] and LLaMA-Adapter [58] are open-source efforts that enable LLaMA to use image inputs, paving the way to build open-source multimodal LLMs. While these models present promising task transfer generalization performance, they are not explicitly tuned with vision-language instruction data, and their performance in multimodal tasks usually falls short compared to language-only tasks. In this paper, we aim to fill this gap and study its effectiveness. Finally, note that visual instruction tuning is different from visual prompt tuning [23]: the former aims to improve the model's instruction-following abilities, while the latter aims to improve the parameter-efficiency in model adaptation.

**Context type 1: Captions**
A group of people standing outside of a black vehicle with various luggage.
Luggage surrounds a vehicle in an underground parking area
People try to fit all of their luggage in an SUV.
The sport utility vehicle is parked in the public garage, being packed for a trip
Some people with luggage near a van that is transporting it.
**Context type 2: Boxes**
person: [0.681, 0.242, 0.774, 0.694], backpack: [0.384, 0.696, 0.485, 0.914], suitcase: ...<omitted>

---

**Response type 1: conversation**
Question: What type of vehicle is featured in the image?
Answer: The image features a black sport utility vehicle (SUV) ...<omitted>
**Response type 2: detailed description**
The image is an underground parking area with a black sport utility vehicle (SUV) parked. There are three people in the scene, with one person standing closer to the left side of the vehicle, another person in the middle, and the third person on the right side. They are all working together to pack their luggage into the SUV for a trip. ...<omitted>
**Response type 3: complex reasoning**
Question: What challenges do these people face?
Answer: In the image, a group of people is standing outside a black SUV in a parking area, surrounded by various pieces of luggage, including suitcases and backpacks. They are facing the challenge of fitting all their luggage into the black SUV. There are multiple suitcases and backpacks to be packed, which suggests that the group has a significant amount of belongings ...<omitted>

Table 1: One example to illustrate the instruction-following data. The top block shows the contexts such as captions and boxes used to prompt GPT, and the bottom block shows the three types of responses. Note that the visual image is not used to prompt GPT, we only show it here as a reference.

# 3 GPT-assisted Visual Instruction Data Generation

The community has witnessed a surge in the amount of public multimodal data such as image-text pairs, ranging from CC [8] to LAION [44]. However, when it comes to multimodal instruction-following data, the available amount is limited, partially because the process for creating such data is time-consuming and less well-defined when human crowd-scouring is considered. Inspired by the success of recent GPT models in text-annotation tasks [17], we propose to leverage ChatGPT/GPT-4 for multimodal instruction-following data collection, based on the widely existing image-pair data.

For an image $\mathbf{X}_v$ and its associated caption $\mathbf{X}_c$, it is natural to create a set of questions $\mathbf{X}_q$ with the intent to instruct the assistant to describe the image content. We prompt GPT-4 to curate such a list of questions (see details in Appendix). Therefore, a simple way to expand an image-text pair to its instruction-following version is `Human : `$\mathbf{X}_q$ $\mathbf{X}_v$`<STOP> Assistant : `$\mathbf{X}_c$`<STOP>`. Though cheap to construct, this simple expanded version lacks diversity and in-depth reasoning in both the instructions and responses.

To mitigate this issue, we leverage language-only GPT-4 or ChatGPT as the strong teacher (both accept only text as input), to create instruction-following data involving visual content. Specifically, in order to encode an image into its visual features to prompt a text-only GPT, we use two types of symbolic representations: (i) *Captions* typically describe the visual scene from various perspectives; (ii) *Bounding boxes* usually localize the objects in the scene, and each box encodes the object concept and its spatial location. One example is shown in the top block of Table 14.

This symbolic representation allows us to encode the image as an LLM-recognizable sequence. We use COCO images [30] and generate three types of instruction-following data. One example per type is shown in the bottom block of Table 14. For each type, we first manually design a few examples. They are the only human annotations we have during data collection, and are used as seed examples in in-context-learning to query GPT-4.

- *Conversation*. We design a conversation between the assistant and a person asking questions about this photo. The answers are in a tone as if the assistant is seeing the image and answering the question. A diverse set of questions are asked about the visual content of the image, including the object types, counting the objects, object actions, object locations, relative positions between

objects. Only questions that have definite answers are considered. Please see Appendix for the detailed prompt.

- *Detailed description.* To include a rich and comprehensive description for an image, we create a list of questions with such an intent. We prompt GPT-4 then curate the list (see detailed prompts and curation process in Appendix). For each image, we randomly sample one question from the list to ask GPT-4 to generate the detailed description.
- *Complex reasoning.* The above two types focus on the visual content itself, based on which we further create in-depth reasoning questions. The answers typically require a step-by-step reasoning process by following rigorous logic.

We collect 158K unique language-image instruction-following samples in total, including 58K in conversations, 23K in detailed description, and 77k in complex reasoning, respectively. We ablated the use of ChatGPT and GPT-4 in our early experiments, and found that GPT-4 consistently provides higher quality instruction-following data, such as spatial reasoning.

## 4 Visual Instruction Tuning

### 4.1 Architecture

The primary goal is to effectively leverage the capabilities of both the pre-trained LLM and visual model. The network archtecture is illustrated in Figure 1. We choose Vicuna [9] as our LLM $f_\phi(\cdot)$ parameterized by $\phi$, as it has the best instruction following capabilities in language tasks among publicly available checkpoints [47, 9, 37].

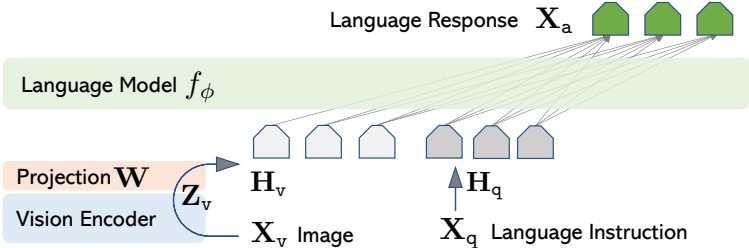

Figure 1: LLaVA network architecture.

For an input image $\mathbf{X}_\mathrm{v}$, we consider the pre-trained CLIP visual encoder ViT-L/14 [39], which provides the visual feature $\mathbf{Z}_\mathrm{v} = g(\mathbf{X}_\mathrm{v})$. The grid features before and after the last Transformer layer are considered in our experiments. We consider a simple linear layer to connect image features into the word embedding space. Specifically, we apply a trainable projection matrix $\mathbf{W}$ to convert $\mathbf{Z}_\mathrm{v}$ into language embedding tokens $\mathbf{H}_\mathrm{v}$, which have the same dimensionality as the word embedding space in the language model:

$$\mathbf{H}_\mathrm{v} = \mathbf{W} \cdot \mathbf{Z}_\mathrm{v}, \text{ with } \mathbf{Z}_\mathrm{v} = g(\mathbf{X}_\mathrm{v}) \tag{1}$$

Thus, we have a sequence of visual tokens $\mathbf{H}_\mathrm{v}$. Note that our simple projection scheme is lightweight, which allows us to iterate data centric experiments quickly. More sophisticated schemes to connect the image and language representations can also be considered, such as gated cross-attention in Flamingo [2] and Q-former in BLIP-2 [27]. We leave exploring possibly more effective and sophisticated architecture designs for LLaVA as future work.

### 4.2 Training

For each image $\mathbf{X}_\mathrm{v}$, we generate multi-turn conversation data $(\mathbf{X}_\mathrm{q}^1, \mathbf{X}_\mathrm{a}^1, \cdots, \mathbf{X}_\mathrm{q}^T, \mathbf{X}_\mathrm{a}^T)$, where $T$ is the total number of turns. We organize them as a sequence, by treating all answers as the assistant's response, and the instruction $\mathbf{X}_\mathrm{instruct}^t$ at the $t$-th turn as:

$$\mathbf{X}_\mathrm{instruct}^t = \begin{cases} \text{Randomly choose } [\mathbf{X}_\mathrm{q}^1, \mathbf{X}_\mathrm{v}] \text{ or } [\mathbf{X}_\mathrm{v}, \mathbf{X}_\mathrm{q}^1], & \text{the first turn } t = 1 \\ \mathbf{X}_\mathrm{q}^t, & \text{the remaining turns } t > 1 \end{cases} \tag{2}$$

This leads to the unified format for the multimodal instruction-following sequence illustrated in Table 2. We perform instruction-tuning of the LLM on the prediction tokens, using its original auto-regressive training objective.

```
Xsystem-message <STOP>
Human : X¹instruct <STOP> Assistant: X¹a <STOP>
Human : X²instruct <STOP> Assistant: X²a <STOP> ···
```

Table 2: The input sequence used to train the model. Only two conversation turns are illustrated here; in practice, the number of turns varies based on the instruction-following data. In our current implementation, we follow Vicuna-v0 [9] to set the system message $\mathbf{X}_{\texttt{system-message}}$ and we set `<STOP>` = ###. The model is trained to predict the assistant answers and where to stop, and thus only green sequence/tokens are used to compute the loss in the auto-regressive model.

Specifically, for a sequence of length $L$, we compute the probability of the target answers $\mathbf{X}_{\texttt{a}}$ by:

$$p(\mathbf{X}_{\texttt{a}}|\mathbf{X}_{\texttt{v}}, \mathbf{X}_{\texttt{instruct}}) = \prod_{i=1}^{L} p_{\boldsymbol{\theta}}(x_i|\mathbf{X}_{\texttt{v}}, \mathbf{X}_{\texttt{instruct},<i}, \mathbf{X}_{\texttt{a},<i}), \qquad (3)$$

where $\boldsymbol{\theta}$ is the trainable parameters, $\mathbf{X}_{\texttt{instruct},<i}$ and $\mathbf{X}_{\texttt{a},<i}$ are the instruction and answer tokens in all turns before the current prediction token $x_i$, respectively. Please see Table 2 for an illustration of the prediction tokens. For the conditionals in (3), we explicitly add $\mathbf{X}_{\texttt{v}}$ to emphasize the fact that the image is grounded for all answers, and we omit $\mathbf{X}_{\texttt{system-message}}$ and all previous `<STOP>` for better readability. For LLaVA model training, we consider a two-stage instruction-tuning procedure.

**Stage 1: Pre-training for Feature Alignment.** To strike a balance between concept coverage and training efficiency, we filter CC3M to 595K image-text pairs. Please see Appendix for details of the filtering process. These pairs are converted to the instruction-following data using the naive expansion method describe in Section 3. Each sample can be treated as a single-turn conversation. To construct the input $\mathbf{X}_{\texttt{instruct}}$ in (2), for an image $\mathbf{X}_{\texttt{v}}$, a question $\mathbf{X}_{\texttt{q}}$ is randomly sampled, which is a language instruction to request the assistant to describe the image briefly. The ground-truth prediction answer $\mathbf{X}_{\texttt{a}}$ is the original caption. In training, we keep both the visual encoder and LLM weights frozen, and maximize the likelihood of (3) with trainable parameters $\boldsymbol{\theta} = \mathbf{W}$ (the projection matrix) only. In this way, the image features $\mathbf{H}_{\texttt{v}}$ can be aligned with the pre-trained LLM word embedding. This stage can be understood as training a compatible visual tokenizer for the frozen LLM.

**Stage 2: Fine-tuning End-to-End.** We always keep the visual encoder weights frozen, and continue to update both the pre-trained weights of the projection layer and LLM in LLaVA; i.e., the trainable parameters are $\boldsymbol{\theta} = \{\mathbf{W}, \boldsymbol{\phi}\}$ in (3). We consider two specific use case scenarios:

- *Multimodal Chatbot.* We develop a Chatbot by fine-tuning on the 158K language-image instruction-following data in Section 3. Among the three types of responses, conversation is multi-turn while the other two are single-turn. They are uniformly sampled in training.

- *Science QA.* We study our method on the ScienceQA benchmark [33], the first large-scale multimodal science question dataset that annotates the answers with detailed lectures and explanations. Each question is provided a context in the form of natural language or an image. The assistant provides the reasoning process in natural language and selects the answer among multiple choices. For training in (2), we organize the data as a single turn conversation, the question & context as $\mathbf{X}_{\texttt{instruct}}$, and reasoning & answer as $\mathbf{X}_{\texttt{a}}$.

## 5 Experiments

We assess the performance of LLaVA in instruction-following and visual reasoning capabilities with two primary experimental settings: multimodal chatbot and the ScienceQA dataset, respectively. We train all models with $8\times$ A100s, following Vicuna's hyperparameters [9]. We pre-train our model on the filtered CC-595K subset for 1 epoch with a learning rate of 2e-3 and a batch size of 128, and fine-tune on the proposed LLaVA-Instruct-158K dataset for 3 epochs, with a learning rate of 2e-5 and a batch size of 32. See Appendix for more training details.

### 5.1 Multimodal Chatbot

We developed a chatbot demo to show the image understanding and conversation abilities of LLaVA, and to study how well LLaVA is able to digest visual inputs and exhibit instruction-following

| | |
|---|---|
| **Visual input example, Extreme Ironing:** | |

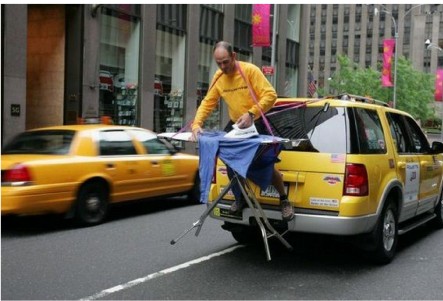

Source: https://www.barnorama.com/wp-content/uploads/2016/12/03-Confusing-Pictures.jpg

| | |
|---|---|
| User | What is unusual about this image? |
| LLaVA | The unusual aspect of this image is a man ironing clothes on the back of a minivan or van. This is not a typical place to perform this activity, as one would usually iron clothes in a more stationary and safe location, such as a home, using a regular ironing board. The scene depicted in the image is peculiar as it involves a makeshift ironing setup on a vehicle, which can be both unsafe and unconventional. Additionally, it is not clear how the man is able to maintain balance and stability while ironing clothes in such an unstable environment. |
| | *[Start a new conversation, and clear the history]* |
| User | What's happening in the scene? |
| LLaVA | The image depicts a man standing on top of a yellow SUV in a busy city street. He is holding a portable ladder, seemingly ironing clothes while standing on the vehicle. Around the scene, there are other cars, a traffic light, and a fire hydrant. Additionally, there are two other people in the vicinity, one close to the man on the car and another further away. The man on the ladder appears to be attracting attention due to his unconventional choice of ironing his clothes on top of a moving car. The city street around him is bustling with activity, adding to the unique nature of the scene. |
| User | What is unusual about this image? |
| GPT-4 [35] | The unusual thing about this image is that a man is ironing clothes on an ironing board attached to the roof of a moving taxi. |
| User | What is unusual about this image? |
| BLIP-2 | a man is sitting on the back of a yellow cab |
| User | What is unusual about this image? |
| OpenFlamingo | The man is drying his clothes on the hood of his car. |

Table 3: Example prompt from GPT-4 paper [35] to compare visual reasoning and chat capabilities. Compared to BLIP-2 [27] and OpenFlamingo [5], LLaVA accurately follows the user's instructions, instead of simply describing the scene. LLaVA offers a more comprehensive response than GPT-4. Even when merely asked to describe the image, LLaVA identifies atypical aspects of the image.

capabilities. We first use the examples in the original GPT-4 paper [35], shown in Table 3 (more examples in Appendix), that require in-depth image understanding. For comparisons, we quote the prompt and response of the multimodal GPT-4 from their paper, and query BLIP-2 and OpenFlamingo model checkpoints to get their response.

Surprisingly, although LLaVA is trained with a small multimodal instruction-following dataset (∼80K unique images), it demonstrates quite similar reasoning results with multimodal GPT-4 on these examples. Note that while these images are out-of-domain for LLaVA, LLaVA is still able to understand the scenes and follow the question instruction to provide a reasonable response. In contrast, BLIP-2 and OpenFlamingo focus on describing the image, instead of following the user instruction to answer in an appropriate manner.

**Quantitative Evaluation.** To gain a systematic understanding of the performance of LLaVA, we propose a quantitative metric to measure the model's instruction-following capability on multimodal data. Inspired by [9], we leverage GPT-4 to measure the quality of generated responses. Specifically, we create triplets consisting of image, ground-truth textual descriptions, and question. The candidate models (*e.g.,* LLaVA) predict the answers based on the question and the image. To provide an *approximate theoretical upper bound*, we create a reference prediction based on the question and the *ground-truth* textual descriptions, using the text-only GPT-4. After obtaining the responses from

|  | Conversation | Detail description | Complex reasoning | **All** |
|---|---|---|---|---|
| Full data | 83.1 | 75.3 | 96.5 | 85.1 |
| Detail + Complex | 81.5 (-1.6) | 73.3 (-2.0) | 90.8 (-5.7) | 81.9 (-3.2) |
| Conv + 5% Detail + 10% Complex | 81.0 (-2.1) | 68.4 (-7.1) | 91.5 (-5.0) | 80.5 (-4.4) |
| Conversation | 76.5 (-6.6) | 59.8 (-16.2) | 84.9 (-12.4) | 73.8 (-11.3) |
| No Instruction Tuning | 22.0 (-61.1) | 24.0 (-51.3) | 18.5 (-78.0) | 21.5 (-63.6) |

Table 4: Ablation on LLaVA-Bench (COCO) with different training data. We report relative scores *w.r.t.* a text-only GPT-4 model that uses ground truth image captions and bounding boxes as visual input. We prompt GPT-4 with the answers from our model outputs and the answers by GPT-4 (text-only), and let it compare between both responses and give a rating with an explanation.

|  | Conversation | Detail description | Complex reasoning | All |
|---|---|---|---|---|
| OpenFlamingo [5] | $19.3 \pm 0.5$ | $19.0 \pm 0.5$ | $19.1 \pm 0.7$ | $19.1 \pm 0.4$ |
| BLIP-2 [27] | $54.6 \pm 1.4$ | $29.1 \pm 1.2$ | $32.9 \pm 0.7$ | $38.1 \pm 1.0$ |
| LLaVA | $57.3 \pm 1.9$ | $52.5 \pm 6.3$ | $81.7 \pm 1.8$ | $67.3 \pm 2.0$ |
| LLaVA$^{\dagger}$ | $58.8 \pm 0.6$ | $49.2 \pm 0.8$ | $81.4 \pm 0.3$ | $66.7 \pm 0.3$ |

Table 5: Instruction-following capability comparison using relative scores on LLaVA-Bench (In-the-Wild). The results are reported in the format of *mean $\pm$ std*. For the first three rows, we report three inference runs. LLaVA performs significantly better than others. $^{\dagger}$ For a given set of LLaVA decoding sequences, we evaluate by querying GPT-4 three times; GPT-4 gives a consistent evaluation.

both models, we feed the question, visual information (in the format of textual descriptions), and the generated responses from both assistants, to the judge (*i.e.,* text-only GPT-4). It evaluates the helpfulness, relevance, accuracy, and level of detail of the responses from the assistants, and gives an overall score on a scale of 1 to 10, where a higher score indicates better overall performance. It is also asked to provide a comprehensive explanation for the evaluation, for us to better understand the models. We report relative scores *w.r.t.* the text-only GPT-4 model that uses the textural ground truth description as visual input. We create two benchmarks to evaluate the model's performance.

**LLaVA-Bench (COCO).** We randomly select 30 images from COCO-Val-2014, and for each image, we generate three types of questions (conversation, detailed description, complex reasoning) using the proposed data generation pipeline in Sec. 3, totaling 90 questions. This benchmark studies the model's alignment behavior and capabilities with consistent visual inputs. We vary the training datasets to study the effectiveness of different types of instruction-following data, and show the results in Table 4. First, with instruction tuning, the model's ability of following user instructions improves significantly by over 50 points. Second, adding a small amount of detailed description and complex reasoning questions contributes to a considerable improvement of the model's overall capability by 7 points. Furthermore, it also improves the model's performance on conversational questions, suggesting that improvements in reasoning capabilities complement conversational abilities. Finally, we show that having all three types of data yields the best performance at 85.1%.

**LLaVA-Bench (In-the-Wild).** To evaluate the model's capability in more challenging tasks and generalizability to novel domains, we collect a diverse set of 24 images with 60 questions in total, including indoor and outdoor scenes, memes, paintings, sketches, *etc.*, and associate each image with a highly-detailed and manually-curated description and a proper selection of questions. We compare LLaVA, BLIP, and OpenFlamingo in Table 5. Thanks to visual instruction tuning, LLaVA achieves significantly better performance compared with BLIP-2 (+29%) and OpenFlamingo (+48%). Compared to the text-only GPT-4 that has access to ground-truth labels, LLaVA achieves an impressive 81.7% performance on complex reasoning questions, with an overall score of 67.3%.

**Limitations.** This LLaVA-Bench (In-the-Wild) is designed to be challenging and to reveal a model's weaknesses. We provide two examples with associated captions and questions in Table 6. For the ramen example (left), to correctly answer the name of the restaurant, it requires the model to have a large knowledge coverage and multilingual understanding capability; to correctly describe the side dishes, the model may need to retrieve relevant multimodal information from Internet. For the fridge example (right), perceiving the correct brand of the yogurt requires the model to process high resolution images and possess extensive knowledge coverage. We also observed an interesting failure of LLaVA, as it responds with *yes* when asked if strawberry-flavored yogurt is present, even though

**Challenging examples from LLaVA-Bench (In-the-Wild):**

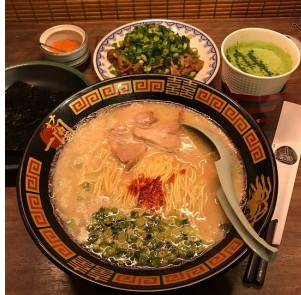 

| | ICHIRAN Ramen [source] | Filled fridge [source] |
| --- | --- | --- |
| Annotation | A close-up photo of a meal at ICHI-RAN. The chashu ramen bowl with a spoon is placed in the center. The ramen is seasoned with chili sauce, chopped scallions, and served with two pieces of chashu. Chopsticks are placed to the right of the bowl, still in their paper wrap, not yet opened. The ramen is also served with nori on the left. On top, from left to right, the following sides are served: a bowl of orange spice (possibly garlic sauce), a plate of smoke-flavored stewed pork with chopped scallions, and a cup of matcha green tea. | An open refrigerator filled with a variety of food items. In the left part of the compartment, towards the front, there is a plastic box of strawberries with a small bag of baby carrots on top. Towards the back, there is a stack of sauce containers. In the middle part of the compartment, towards the front, there is a green plastic box, and there is an unidentified plastic bag placed on it. Towards the back, there is a carton of milk. In the right part of the compartment, towards the front, there is a box of blueberries with three yogurts stacked on top. The large bottle of yogurt is Fage non-fat yogurt, and one of the smaller cups is Fage blueberry yogurt. The brand and flavor of the other smaller cup are unknown. Towards the back, there is a container with an unknown content. |
| Question 1 | What's the name of the restaurant? | What is the brand of the blueberry-flavored yogurt? |
| Question 2 | Describe this photo in detail. | Is there strawberry-flavored yogurt in the fridge? |

Table 6: Challenging examples from LLaVA-Bench (In-the-Wild), we provide extremely-detailed annotation for each image for an accurate evaluation. Some questions require the model to extract details from high resolution image and to have a broad knowledge coverage.

the fridge contains only yogurt *and* strawberries. This indicates that, at times, LLaVA perceives the image as a "bag of patches", failing to grasp the complex semantics within the image. We hope LLaVA serves as a solid baseline on the benchmarks, on which our findings can inspire future work in developing more capable LMMs.

## 5.2 ScienceQA

ScienceQA [33] contains 21k multimodal multiple choice questions with rich domain diversity across 3 subjects, 26 topics, 127 categories, and 379 skills. The benchmark dataset is split into training, validation, and test splits with 12726, 4241, and 4241 examples, respectively. We consider two representative methods, including GPT-3.5 model (text-davinci-002) with and without chain-of-thought (CoT), LLaMA-Adapter [58], as well as multimodal chain-of-thought (MM-CoT) [60], which is the current SoTA method on this dataset. For more baseline numbers, please see [33].

The results are reported in Table 7. For LLaVA, we use the visual features before the last layer, ask the model to first predict reasons and then the answer, and train it for 12 epochs. It yields 90.92% accuracy, which is quite close to the SoTA 91.68%. To explore the limit of LLMs, we also prompt GPT-4 using 2-shot in-context-learning and achieve 82.69% accuracy, which is a 7.52% absolute gain compared with 75.17% from GPT-3.5. For a substantial number of questions, we note that GPT-4 fails simply because it reports that there is insufficient context such as images or plots. We consider two schemes to combine the outcomes from our model and GPT-4. (*i*) *A GPT-4 complement*. Whenever GPT-4 fails to provide answers, we use the prediction from our method. This schemes yields 90.97% accuracy, which is almost the same as applying our method alone. (*ii*) *GPT-4 as the judge*. Whenever GPT-4 and LLaVA produce different answers, we prompt GPT-4 again, asking it to provide its own final answer based on the question and two outcomes. The spirit is similar with CoT, but with the external knowledge from the other model. Surprisingly, this scheme is able to provide consistent improvement over all question classes, and achieves a new SoTA accuracy of 92.53%. Interestingly,

| Method | Subject | | | Context Modality | | | Grade | | Average |
|---|---|---|---|---|---|---|---|---|---|
| | NAT | SOC | LAN | TXT | IMG | NO | G1-6 | G7-12 | |
| *Representative & SoTA methods with numbers reported in the literature* | | | | | | | | | |
| Human [33] | 90.23 | 84.97 | 87.48 | 89.60 | 87.50 | 88.10 | 91.59 | 82.42 | 88.40 |
| GPT-3.5 [33] | 74.64 | 69.74 | 76.00 | 74.44 | 67.28 | 77.42 | 76.80 | 68.89 | 73.97 |
| GPT-3.5 w/ CoT [33] | 75.44 | 70.87 | 78.09 | 74.68 | 67.43 | 79.93 | 78.23 | 69.68 | 75.17 |
| LLaMA-Adapter [58] | 84.37 | 88.30 | 84.36 | 83.72 | 80.32 | 86.90 | 85.83 | 84.05 | 85.19 |
| MM-CoT$_{Base}$ [60] | 87.52 | 77.17 | 85.82 | 87.88 | 82.90 | 86.83 | 84.65 | 85.37 | 84.91 |
| MM-CoT$_{Large}$ [60] | 95.91 | 82.00 | 90.82 | 95.26 | 88.80 | 92.89 | 92.44 | 90.31 | 91.68 |
| *Results with our own experiment runs* | | | | | | | | | |
| GPT-4$^\dagger$ | 84.06 | 73.45 | 87.36 | 81.87 | 70.75 | 90.73 | 84.69 | 79.10 | 82.69 |
| LLaVA | 90.36 | 95.95 | 88.00 | 89.49 | 88.00 | 90.66 | 90.93 | 90.90 | 90.92 |
| LLaVA+GPT-4$^\dagger$ (complement) | 90.36 | 95.50 | 88.55 | 89.05 | 87.80 | 91.08 | 92.22 | 88.73 | 90.97 |
| LLaVA+GPT-4$^\dagger$ (judge) | 91.56 | 96.74 | 91.09 | 90.62 | 88.99 | 93.52 | 92.73 | 92.16 | **92.53** |

Table 7: Accuracy (%) on Science QA dataset. Question categories: NAT = natural science, SOC = social science, LAN = language science, TXT = text context, IMG = image context, NO = no context, G1-6 = grades 1-6, G7-12 = grades 7-12. $^\dagger$Text-only GPT-4, our eval. Our novel model ensembling with the text-only GPT-4 consistently improves the model's performance under all categories, setting the new SoTA performance.

the text-only GPT-4, which cannot process images, improves the overall performance of the model on questions that have an image as context. This is because some of these questions do not actually require the image context for a correct answer. The GPT-4 judge can identify such cases and correct some of the errors that LLaVA makes. See the example in Appendix. To the best of our knowledge, this is the first time that GPT-4 is used for model ensembling. We hope this finding can encourage future research to explore more effective methods to leverage LLMs for model ensembling.

**Ablations.** We ablate several design choices on ScienceQA in Table 8. $(i)$ *Visual features*. We tried using the last layer feature from CLIP vision encoder, which yields 89.96% and is 0.96% lower than the feature before the last layer. We hypothesize that this is because CLIP's last layer features may focus more on global and abstract image properties compared to the layer before it, which can focus more on localized properties that are useful for under-

| Visual features | Before | Last |
|---|---|---|
| Best variant | 90.92 | 89.96 (-0.96) |
| Predict answer first | - | 89.77 (-1.15) |
| Training from scratch | 85.81 (-5.11) | - |
| 7B model size | 89.84 (-1.08) | - |

Table 8: Design choice ablations (%). The difference with the best variant is reported in red text.

standing specific image details. $(ii)$ *Chain-of-thought*. To decide the order between the answer and reasoning process in the model prediction, we run both variants and observe that answer-first reports the best number 89.77% accuracy in 12 epochs, while reasoning-first can quickly reach 89.77% accuracy in 6 epochs, but no further improvement with more training. Training the model for 24 epochs does not improve the performance. We conclude that CoT-like reasoning-first strategy can largely improve convergence, but contributes relatively little to the final performance. $(iii)$ *Pre-training*. We skip pre-training and directly train on Science QA from scratch – performance drops to 85.81% accuracy. The 5.11% absolute degradation indicates the importance of our pre-training stage, in aligning multimodal features while preserving the vast pre-trained knowledge. $(iv)$ *Model size*. We keep all configurations the same as our best 13B model, and train a 7B model. This yields 89.84% accuracy, which is 1.08% lower than 90.92%, demonstrating the importance of model scale.

# 6 Conclusion

This paper demonstrated the effectiveness of visual instruction tuning. We presented an automatic pipeline to create language-image instruction-following data, based on which we train LLaVA, a multimodal model to follow human intent to complete visual tasks. It achieves the new SoTA accuracy when fine-tuned on ScienceQA, and excellent visual chat capabilities when fine-tuned on multimodal chat data. Besides, we present the first benchmark to study multimodal instruction-following capability. This paper is an initial step in visual instruction tuning, and mainly focuses on real-life tasks. For more quantitative results of LLaVA on academic benchmarks, please refer to the

improved baselines with visual instruction tuning [31]. We hope our work can inspire future research on building more capable multimodal models.

**Acknowledgements.** We thank Baolin Peng and Pan Lu for valuable discussions on instruction-tuning language models and Science QA, respectively. We thank the LLaMA team for giving us access to their models, and open-source projects, including Alpaca and Vicuna. This work was supported in part by NSF CAREER IIS2150012, and Institute of Information & communications Technology Planning & Evaluation(IITP) grants funded by the Korea government(MSIT) (No. 2022-0-00871, Development of AI Autonomy and Knowledge Enhancement for AI Agent Collaboration) and (No. RS-2022-00187238, Development of Large Korean Language Model Technology for Efficient Pre-training).

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

# A    Broader Impact

The broader impact of LLaVA, a general-purpose visual assistant, has potential benefits and risks associated with its deployment and release. Some considerations are unique to LLaVA due to its visual nature, while others share similarities with existing instruction-following LLMs (*e.g.,* Alpaca, Vicuna, *etc.*). As LLaVA is built upon LLaMA, Vicuna, and CLIP, it inherits some of the issues associated with LLMs and vision encoders. In the following, we outline both the risks and mitigation strategies in place for the release of this model.

**Malicious input.**    To minimize potential misuse and harmful consequences, we employ two precautionary measures for LLaVA: (1) *OpenAI Filter API* for user input text to prevent harmful or inappropriate text instructions from being processed by the model, and (2) *NSFW Filter* for uploaded user images to detect and block Not Safe For Work (NSFW) content or any other potentially harmful image inputs.

**Hallucination.**    Similar to LLMs, LLaVA might generate outputs that aren't grounded in facts or input data. This raises concerns about inferences made, especially in critical applications (*e.g.,* medical).

**Biases.**    Bias can be transferred from the base models to LLaVA, both from the vision encoder (CLIP) and the language decoder (LLaMA/Vicuna). This may lead to biased outcomes or unfair representations of diverse content.

**Energy consumption.**    Though energy consumption is not a primary concern for LLaVA due to a smaller pretraining dataset (see details in Sec. C), it may become a concern when scaling up the pretraining dataset or increasing the model size, e.g., to a larger LLaMA version like the 65B model.

**Evaluation complexities.**    Assessing the performance of LLaVA is challenging as it involves both language and visual tasks. Our evaluation benchmark covers several aspects, including accuracy, concept coverage, reasoning ability, and creativity. However, additional aspects need consideration, such as the degree of visual content hallucination and fine-grained understanding of visual content. While text-only GPT-4 based multimodal evaluation is consistent and accurate in our study, its robustness in different situations and capability to evaluate other unexplored aspects are subjects for future work.

Despite these risks, we believe that the benefits of releasing LLaVA to the research community outweigh the potential harm. It allows for ongoing investigation and improvement of the model and engages the community in developing better mitigation strategies to address these concerns. Moreover, the release of LLaVA can spur the development of new applications and research directions, ultimately contributing to the progress and responsible deployment of foundation models in vision-language tasks.

# B    More Results

We present more qualitative results of LLaVA to analyze its emergent behaviors and observed weaknesses. For more quantitative results of LLaVA on academic benchmarks, please refer to the improved baselines with visual instruction tuning [31]. In Table 9, LLaVA demonstrates a similar behavior as GPT-4 in another example from its paper. Similar to the GPT-4 live demo by OpenAI, LLaVA is capable of generating the HTML/JS/CSS code for an interactive joke website based on a simplified user input sketch in Fig. 2, despite a minor error. As shown in Fig. 3, LLaVA can follow user's instructions in a conversational style and provide detailed responses or creative writings. Furthermore, LLaVA is able to relate the visual content to the textual knowledge from the pretrained LLM, as demonstrated in Fig. 4 and Fig. 5.

One interesting emergent behavior of LLaVA is that it is able to understand visual contents that are not covered in the training. For example, in Fig. 6, it is able to recognize Elon Musk both in a headshot and in a humorous meme where he is dressed as a doge, even though Elon Musk *never* appears in the training data for either the visual feature alignment or visual instruction tuning stages of

LLaVA. LLaVA also demonstrates impressive OCR (optical character recognition) ability in Table 9 and Fig. 2, which is rarely covered in our training data.

We hope these additional results and observations showcase the potential of LLaVA in various application areas. In future work, it is important to investigate these emergent behaviors more thoroughly and to understand the underlying mechanisms that enable LLaVA to demonstrate such generalization abilities. This will pave the way towards building better LMMs, including enhancing robustness, reducing biases, and improving the alignment and the scope of the learned vision-language representations.

---

**Visual input example, Chicken Nugget Map:**

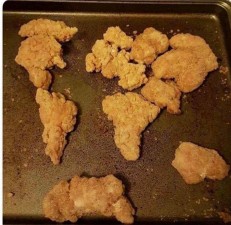

Sometimes I just look at pictures of the earth from space and I marvel at how beautiful it all is.

| | |
|---|---|
| User | Can you explain this meme in detail? |
| LLaVA | The meme in the image is a creative and humorous take on food, with a focus on chicken nuggets, as the center of the universe. The meme begins with the phrase "Sometimes I just look at pictures of the Earth from space and I marvel at how beautiful it all is..." The main image features a tray of cooked chicken nuggets, with the chicken pieces positioned to resemble the continents and islands on a world map. The meme playfully suggests that the chicken nuggets represent the Earth, and the various locations depicted in the photo are actually chicken nugget versions of different places. The punchline of the meme appears in smaller text at the bottom, reading "I mean, it's not the real Earth, but how beautiful it is all is." This lighthearted and entertaining meme combines food and humor, using the image of chicken nuggets to create a funny and imaginative representation of the world. |
| User | Can you explain this meme? |
| GPT-4 [35] | This meme is a joke that combines two unrelated things: pictures of the earth from space and chicken nuggets. The text of the meme suggests that the image below is a beautiful picture of the earth from space. However, the image is actually of chicken nuggets arranged to vaguely resemble a map of the world. The humor in this meme comes from the unexpected juxtaposition of the text and the image. The text sets up an expectation of a majestic image of the earth, but the image is actually something mundane and silly. |
| User | Can you explain this meme in detail? |
| BLIP-2 | sometimes i just look at pictures of the earth from space and marvel how beautiful it is |
| User | Can you explain this meme in detail? |
| OpenFlamingo | It's a picture of a chicken nugget on the International Space Station. |

Table 9: Example prompt comparing LLaVA, GPT-4, BLIP-2, and OpenFlamingo's visual reasoning capabilities in understanding the humor. BLIP-2 and OpenFlamingo fail to follow the user's instructions. LLaVA and GPT-4 both explain the meme and its humor, while GPT-4 produces a more concise answer.

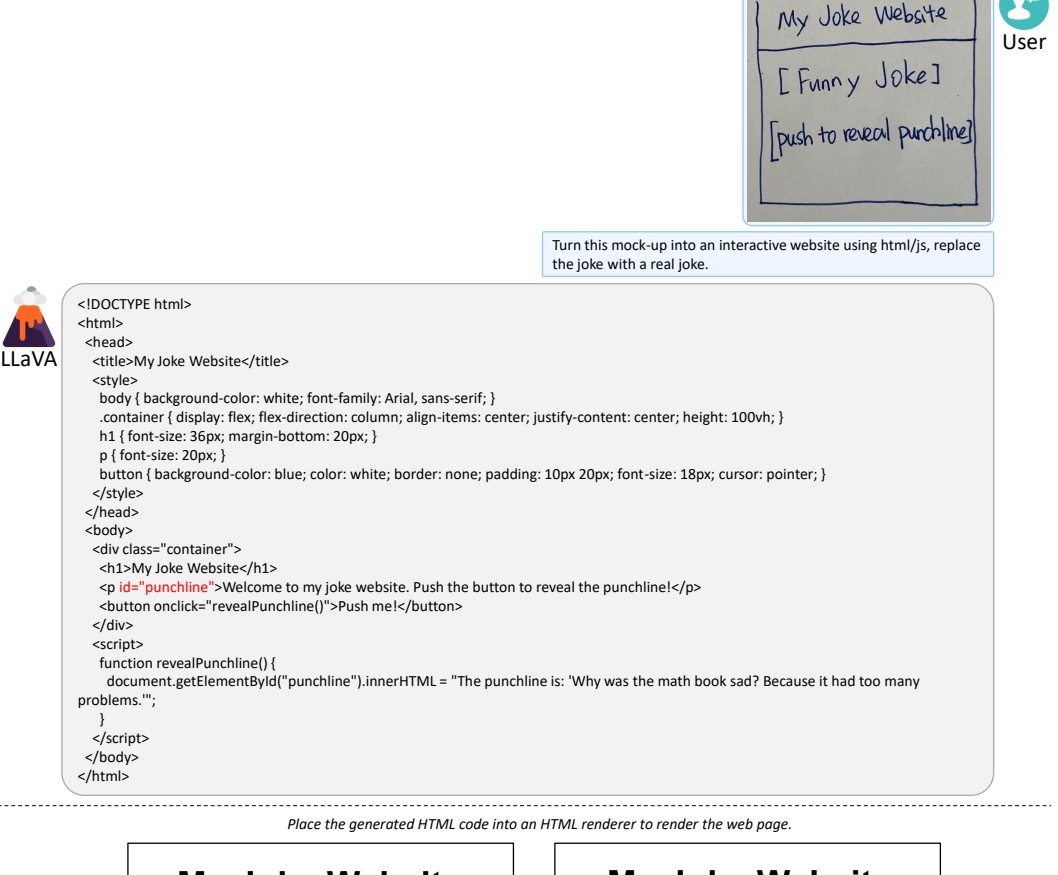

Figure 2: LLaVA generates HTML/JS code for an interactive website based on user sketch inputs. The interactive interface works after fixing a minor error (*in red*) in the generated output. There is room for improvement in LLaVA's output, such as splitting the joke and punchline into two rows, and only revealing the punchline upon button click, to better reflect the user's intent.

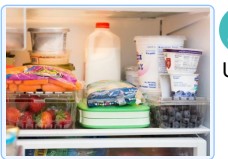

**User**

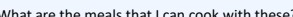

What are the meals that I can cook with these?

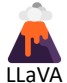

**LLaVA**

With the variety of food items stored in the refrigerator, you can prepare several meals. For example, you can create a fruit salad using the strawberries, blueberries, and carrots. Additionally, you can make a delicious smoothie using the fruits and vegetables, which would be a healthy and nutritious snack or breakfast option. The milk and eggs can be used to create various dishes, such as a milk-based smoothie, a baked casserole, or scrambled eggs. The juice can also be used to make a fresh fruit and juice mix or as a refreshing homemade popsicle. The possibilities are vast, and the meals you can create will depend on your preferences and dietary needs.

I would like to do the fruit salad. Show me the recipe.

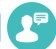 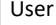

**User**

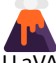

**LLaVA**

Certainly! Here's a simple fruit salad recipe using the items in your refrigerator:

Ingredients:
- 2 cups fresh strawberries, sliced
- 1 cup fresh blueberries
- 1 large carrot, grated
- Juice of half a lemon
- A handful of chopped fresh parsley or mint
- A few ice cubes
- Optional: honey or maple syrup (adjust to taste)

Instructions:
1. In a large bowl, combine the sliced strawberries, blueberries, and carrot. Gently toss them together to mix.
2. Drizzle the juice of half a lemon over the fruit and vegetables to prevent them from oxidizing and add a refreshing citrus flavor.
3. Sprinkle the chopped fresh parsley or mint over the fruit and gently toss once more.
4. If desired, add a drizzle of honey or maple syrup for a touch of sweetness. Adjust the amount according to your taste preferences.
5. Cover the bowl and refrigerate the fruit salad for at least 30 minutes or up to a few hours. This allows the flavors to meld and creates a more refreshing salad.
6. Serve the fruit salad in cups or bowls, garnished with a sprinkle of fresh parsley or mint, if desired.

This fruit salad recipe is perfect for a healthy snack or side dish, and with the variety of ingredients available in your refrigerator, it should be easy to put together quickly.

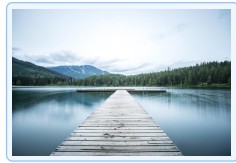

**User**

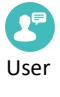

What are the things I should be cautious about when I visit here?

**LLaVA**

When visiting this peaceful location with a pier extending over a serene lake, surrounded by mountains, there are several things to be cautious about. First, pay attention to the weather conditions, as sudden changes in weather, especially strong winds or storms, can cause the water to become rough and pose risks to the stability of the pier. Second, be mindful of the structural integrity of the pier; avoid overloading it with too many people or excessive weight, as this could potentially damage the pier or cause it to collapse. Additionally, be aware of the surrounding wildlife and respect their space; do not harm or disturb them in any way. Finally, if you decide to go fishing, ensure that you adhere to any local fishing regulations and use appropriate fishing techniques to keep the lake ecosystem healthy and maintain the balance of the food chain.

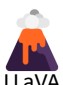

Compose an engaging travel blog post about my trip to this place, highlighting the beautiful sceneries, joyful experiences, and include possible anecdotes.

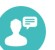

**User**

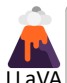

**LLaVA**

Traveling to this picturesque location with a serene lake and mountains in the background was a refreshing and rejuvenating experience. The pier extending over the calm water provided an excellent vantage point from which to appreciate the breathtaking scenery. As I stood on the pier, I couldn't help but feel a deep connection with nature, taking a moment to soak in the peaceful atmosphere.

One of the highlights of my trip was undoubtedly the fishing experience. The area is teeming with life, and I was thrilled to land a sizeable catch, despite the modest size of the lake. The joy of fishing in such a beautiful setting was truly rewarding, and it allowed me to take a break from my busy life and truly immerse myself in the tranquility of nature.

While exploring the surrounding area, I came across a family of birds that had built a nest close to the pier, adding to the charm of the location. Interacting with the wildlife and observing their behavior was an enriching experience that I'll cherish for years to come.

In conclusion, this idyllic destination provided me with an opportunity to unwind and recharge, surrounded by mesmerizing natural beauty and the gentle sounds of water and wildlife. If you're looking for a serene escape from the hustle and bustle of everyday life, this pier overlooking the lake is the perfect place to find solace and reconnect with nature.

Figure 3: LLaVA is capable of recognizing the visual content following the user's intent, without directly prompting for visual recognition. It also provides a detailed response when prompted with a follow-up request, and the generated response is closely related to the provided visual content.

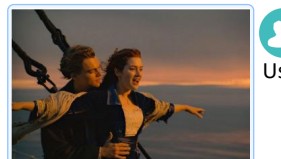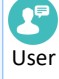
User

What's the ending of this movie?

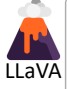
LLaVA

In the image, a young woman is leaning on a man in front of a beautiful sunset, with both of them standing on a boat. They appear to be enjoying a romantic moment together. The scene suggests that they could be depicted as the characters Jack and Rose from the popular movie Titanic. In the movie, their romance blossoms aboard the fictional ocean liner Titanic, with the pair ultimately sharing a passionate kiss as the ship sinks into the ocean. The ending of the movie is known for its emotional and heartfelt scenes, where the characters are forced to say goodbye to each other as they are separated by the sinking ship. The image evokes a sense of romance and nostalgia, paying tribute to the iconic love story portrayed in the movie Titanic.

Figure 4: LLaVA relates the movie scenes to the textual knowledge from the pretrained LLM.

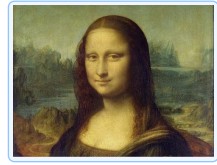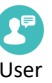
User

Do you know who drew this painting?

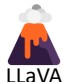
LLaVA

The painting depicts a woman, commonly believed to be Mona Lisa, the famous artwork by Leonardo da Vinci. It is a portrait painting that showcases the woman's enigmatic smile and has become one of the most famous and iconic art pieces in the world. The original work is displayed in the Louvre Museum in Paris, and it is known for its intricate details, use of oil paint, and the artist's innovative techniques that contributed to its enduring appeal and mystery.

*Start a new conversation, and the history is cleared.*

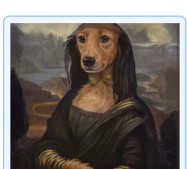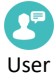
User

Do you know who drew this painting?

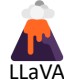
LLaVA

The painting depicts a dog in a humorous situation, where it appears to be wearing a woman's dress and standing in front of a landscape. Although the artist of the painting is not explicitly identified, it could be a reference to the famous artist Da Vinci, known for his artistic works, including the Mona Lisa. The painting is likely a creative and comical take on the traditional portrait style, replacing a human figure with a dog dressed in a woman's clothing, making it an interesting and amusing visual piece.

Figure 5: LLaVA recognizes the famous art work, Mona Lisa, by Leonardo da Vinci. When we start a new conversation, it also explains the humourous artwork created on the web, mimicking the Mona Lisa.

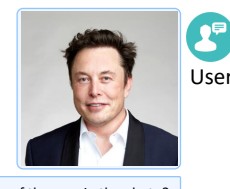

User

What is the name of the man in the photo?

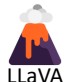

LLaVA The name of the man in the photo is Elon Musk.

*Start a new conversation, and the history is cleared.*

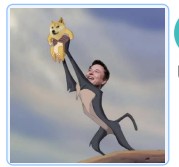

User

What is the name of the man in the photo?

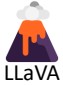

LLaVA The name of the man in the photo is Elon Musk.

Figure 6: An interesting emergent behavior of LLaVA is its ability to recognize Elon Musk both in a headshot and in a humorous meme where he is dressed as a doge. This implies that the pre-trained CLIP vision encoder may have seen images of Elon Musk. However, it is still surprising because Elon Musk *never* appears in the training data for either the visual feature alignment or visual instruction tuning stages of LLaVA, which indicates that the base language model generalizes to unseen visual concepts.

---

**Question:**
Which material is this rocking chair made of?
Options: (A) wood (B) silk

**LLaVA answer:**
LECTURE: A material is a type of matter.
Wood, glass, metal, and plastic are common materials.
Some objects are made of just one material.
Most nails are made of metal.
Other objects are made of more than one material.
This hammer is made of metal and wood.
SOLUTION: Look at the picture of the rocking chair. The rocking chair is made of two different materials. The legs are made of wood, and the back and seat are made of silk.
The answer is B.

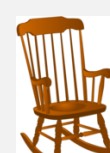

**GPT-4 (text-only) answer:**
The answer is A. BECAUSE: Rocking chairs are typically made of sturdy materials that can support the weight of a person and withstand the rocking motion. Wood is a common material used for making rocking chairs due to its strength, durability, and natural appearance.

---

**GPT-4 (text-only) judge:**
Since there is no context or image to refer to, I cannot independently analyze the rocking chair in question. However, I can try to analyze the reasoning given by the assistants.
Assistant 1, who claims to have seen the image, mentions that the legs of the rocking chair are made of wood, and the back and seat are made of silk, implying that it's a combination of two materials. However, this response seems less plausible, as silk is not a common material used for the construction of rocking chairs due to issues with stability and durability. Typically, one would expect the frame, legs, and seat or back to be made of wood, with cushions and padding made of fabrics, such as cotton or polyester.
Assistant 2 argues that rocking chairs are typically made of sturdy materials, and wood is a common material used for making rocking chairs due to its strength, durability, and natural appearance. This reasoning stands true and is based on common knowledge.
Given the information available, the most plausible answer is that the rocking chair is made of wood, as suggested by Assistant 1. The answer is A.

Table 10: One example on how the text-only GPT-4 acts as a judge to ensemble the predictions from LLaVA and a text-only GPT-4, and gives a correct final answer.

## C   Training Details

We pre-train our model on the filtered CC-595K subset for 1 epoch with a learning rate of 2e-3 and a batch size of 128, and fine-tune on the proposed LLaVA-Instruct-158K dataset for 3 epochs, with a learning rate of 2e-5 and a batch size of 32. Following Vicuna, we use the Adam optimizer with no weight decay and a cosine learning rate with a warmup ratio of 3%. During finetuning, FSDP (Full Shard Data Parallel) and gradient checkpointing is used to save GPU memory, and offloading is not used. BF16 and TF32 are enabled to achieve a balance between speed and precision.

We train all models with $8\times$ A100s. Pretraining on CC-595K completes within 4 hours. Finetuning on Instruct-158K completes within 10 hours. Finetuning on ScienceQA completes within 4 hours.

## D   Assets

Our source code, generated instruction-tuning data, proposed benchmark are uploaded to the anonymized GitHub repository: LLaVA-Annonymous/LLaVA.

1. Source Code: link
2. README: link
3. Instructions to launch the demo: link
4. All prompts and few shot examples for querying GPT-4: link
5. LLaVA-Instruct-158K: link
6. LLaVA-Bench: COCO, In-The-Wild
7. Model checkpoints. The size of the model checkpoints after compression is 25GB, which exceeds the 5GB limit of GitHub LFS (Large File Storage). We'll release the checkpoint to the public, or upon request with reviewers for this submission.

## E   Data

**Instructions for brief image description.**   The list of instructions used to briefly describe the image content are shown in Table 11. They present the same meaning with natural language variance.

> - "Describe the image concisely."
> - "Provide a brief description of the given image."
> - "Offer a succinct explanation of the picture presented."
> - "Summarize the visual content of the image."
> - "Give a short and clear explanation of the subsequent image."
> - "Share a concise interpretation of the image provided."
> - "Present a compact description of the photo's key features."
> - "Relay a brief, clear account of the picture shown."
> - "Render a clear and concise summary of the photo."
> - "Write a terse but informative summary of the picture."
> - "Create a compact narrative representing the image presented."

Table 11: The list of instructions for brief image description.

**Instructions for detailed image description.**   The list of instructions used to describe the image content in detail are shown in Table 12. They present the same meaning with natural language variance.

**CC3M.**   We extract noun-phrases using Spacy for each caption over the whole CC3M dataset, and count the frequency of each unique noun-phrase. We skip noun-phrases whose frequency is smaller than 3, as they are usually rare combinations concept and attributes that has already been covered

- "Describe the following image in detail"
- "Provide a detailed description of the given image"
- "Give an elaborate explanation of the image you see"
- "Share a comprehensive rundown of the presented image"
- "Offer a thorough analysis of the image"
- "Explain the various aspects of the image before you"
- "Clarify the contents of the displayed image with great detail"
- "Characterize the image using a well-detailed description"
- "Break down the elements of the image in a detailed manner"
- "Walk through the important details of the image"
- "Portray the image with a rich, descriptive narrative"
- "Narrate the contents of the image with precision"
- "Analyze the image in a comprehensive and detailed manner"
- "Illustrate the image through a descriptive explanation"
- "Examine the image closely and share its details"
- "Write an exhaustive depiction of the given image"

Table 12: The list of instructions for detailed image description.

by other captions. We then start from the noun-phrases with lowest remaining frequency, add the captions that contain this noun-phrase to the candidate pool. If the frequency of the noun-phrase is larger than 100, we randomly choose a subset of size 100 out of all its captions. This results in around 595K image-text pairs.

The comparison of noun-phrase statistics before and after filtering CC3M is shown in Figure 7. The filtered dataset shows a good coverage of concepts whose frequency is higher from 3, but with a smaller number of image-text pairs.

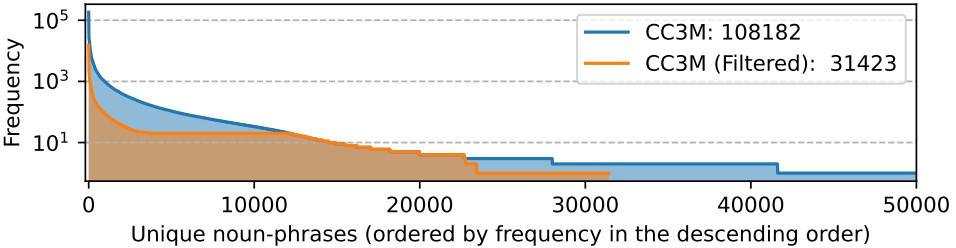

Figure 7: Comparison of noun-phrase statistics before and after filtering CC3M. The total number of unique noun-phrases are reported in the legend.

## F   Prompts

The prompt used to generate image-based conversation from ChatGPT/GPT-4 is shown in Table 13.

```
messages = [ {"role":"system", "content": f"""You are an AI visual assistant, and you are
seeing a single image. What you see are provided with five sentences, describing the same image you
are looking at. Answer all questions as you are seeing the image.

Design a conversation between you and a person asking about this photo. The answers should be in a
tone that a visual AI assistant is seeing the image and answering the question. Ask diverse questions
and give corresponding answers.

Include questions asking about the visual content of the image, including the object types, counting
the objects, object actions, object locations, relative positions between objects, etc. Only include
questions that have definite answers:
(1) one can see the content in the image that the question asks about and can answer confidently;
(2) one can determine confidently from the image that it is not in the image. Do not ask any question
that cannot be answered confidently.

Also include complex questions that are relevant to the content in the image, for example, asking
about background knowledge of the objects in the image, asking to discuss about events happening in
the image, etc. Again, do not ask about uncertain details. Provide detailed answers when answering
complex questions. For example, give detailed examples or reasoning steps to make the content more
convincing and well-organized. You can include multiple paragraphs if necessary."""}
]
for  sample in fewshot_samples:
    messages.append({"role":"user", "content":sample['context']})
    messages.append({"role":"assistant", "content":sample['response']} )
messages.append({"role":"user", "content":'\n'.join(query)})
```

Table 13: For each query, we illustrate the prompt construction process for ChatGPT/GPT-4 to
collect `query['response']` from `query['context']`, using few-shot in-context-learning, where
examples are from `fewshot_samples`, each example including input `sample['context']` and
output `sample['response']`. Note that `messages` is the final prompt. In this example, we provide
the prompt used to generate the conversation response, please see also see its in-context-learning
examples in Table 15 and Table 16 for details. We recommend readers to check out the codebase for
the prompts to generated two other types of responses, including detailed decription and complex
reasoning.

**Context type 1: Captions**
A group of people standing outside of a black vehicle with various luggage.
Luggage surrounds a vehicle in an underground parking area
People try to fit all of their luggage in an SUV.
The sport utility vehicle is parked in the public garage, being packed for a trip
Some people with luggage near a van that is transporting it.

**Context type 2: Boxes**
person: [0.681, 0.242, 0.774, 0.694], person: [0.63, 0.222, 0.686, 0.516], person: [0.444, 0.233, 0.487, 0.34], backpack: [0.384, 0.696, 0.485, 0.914], backpack: [0.755, 0.413, 0.846, 0.692], suitcase: [0.758, 0.413, 0.845, 0.69], suitcase: [0.1, 0.497, 0.173, 0.579], bicycle: [0.282, 0.363, 0.327, 0.442], car: [0.786, 0.25, 0.848, 0.322], car: [0.783, 0.27, 0.827, 0.335], car: [0.86, 0.254, 0.891, 0.3], car: [0.261, 0.101, 0.787, 0.626]

---

**Response type 1: conversation**
Question: What type of vehicle is featured in the image?
Answer: The image features a black sport utility vehicle (SUV).
Question: Where is the vehicle parked?
Answer: The vehicle is parked in an underground parking area, likely in a public garage.
Question: What are the people in the image doing?
Answer: The people are trying to fit all of their luggage into the SUV, likely preparing for a trip.

**Response type 2: detailed description**
The image is an underground parking area with a black sport utility vehicle (SUV) parked. There are three people in the scene, with one person standing closer to the left side of the vehicle, another person in the middle, and the third person on the right side. They are all working together to pack their luggage into the SUV for a trip.

In the parking area, various luggage items are scattered around the vehicle. There are two backpacks, one located near the left rear wheel and the other closer to the right side of the vehicle. Additionally, there are two suitcases, one on the right side of the car and another further away near the center of the parking area. A bicycle can also be seen on the left side of the vehicle.

Other cars are parked around the main SUV, with one car positioned behind it and slightly to the left, another behind and slightly to the right, and the third car further behind on the right side.

**Response type 3: complex reasoning**
Question: What challenges do these people face?
Answer: In the image, a group of people is standing outside a black SUV in a parking area, surrounded by various pieces of luggage, including suitcases and backpacks. They are facing the challenge of fitting all their luggage into the black SUV. There are multiple suitcases and backpacks to be packed, which suggests that the group has a significant amount of belongings to accommodate. They might have to strategize and arrange the luggage efficiently to ensure that everything fits properly into the vehicle. Additionally, they need to consider the comfort of the passengers and visibility while driving, so the placement of the luggage must not obstruct the driver's view or make the passengers uncomfortable during the trip.

Table 14: One example to illustrate the instruction-following data. The top block shows the contexts such as captions and boxes used to prompt GPT, and the bottom block shows the three types of responses. Note that the visual image is not used to prompt GPT, we only show it here as a reference.

**Captions**
There is a movie theater that displays the show times above the doors.
A red fire hydrant is deep in the snow.
The fire hydrant is in the snow near a recently plowed sidewalk.
This city has had a very hard winter with snow.
A hotel for dogs in the snow in winter.

**Conversation**
Question:
What color is the fire hydrant in the image?
===
Answer:
The fire hydrant in the image is red.
===
Question:
Is the fire hydrant fully covered by snow or partially visible?
===
Answer:
The fire hydrant is partially visible, with a significant portion of it covered by the deep snow.
===
Question:
What is the condition of the sidewalk near the fire hydrant?
===
Answer:
The sidewalk near the fire hydrant has been recently plowed, so it is relatively clear of snow compared to the surrounding area.
===
Question:
What does the image suggest about the winter season in the city?
===
Answer:
The image suggests that the winter season in the city has been quite harsh and challenging. The significant snow accumulation seen in the image indicates that the city has experienced heavy snowfall, which can lead to various difficulties and disruptions for its residents.

For instance, the deep snow covering the fire hydrant could potentially hinder its accessibility in case of emergencies, which poses safety concerns. The recently plowed sidewalk implies that the city's maintenance crews have been actively working to keep the walkways clear and safe for pedestrians, but the sheer amount of snowfall might make it difficult to keep up with the continuous snow removal efforts.

Furthermore, such extreme winter conditions can impact transportation, leading to traffic issues, delays in public transit, and increased risks of accidents due to slippery roads. It can also cause problems for businesses and schools, as people might face challenges commuting to work or attending classes. Additionally, the heavy snow can put extra strain on infrastructure, such as roofs and power lines, increasing the likelihood of structural damage or power outages.

In conclusion, the image of the red fire hydrant deep in the snow and the recently plowed sidewalk suggest that the city has faced a particularly severe winter season, with substantial snowfall that has likely caused various challenges and disruptions for its residents and infrastructure.

Table 15: One example used in in-context-learning to construct visual conversation data.

**Captions**

A man is skiing in the open snow covered hills
A skier is making his way into the snow.
A skier on the foothills of a huge mountain range.
A skier looks at mountains while standing near a trail sign.
a single person skiing by an area with a lot of bushes

---

**Conversation**

Question:
What is the skier doing?
===
Answer:
The skier is skiing in the snow-covered hills, making his way through the snow and enjoying the beautiful mountain scenery. They are also standing near a trail sign, which indicates their location on the mountain.
===
Question:
Are there any other people or skiers visible in the image?
===
Answer:
No, there is only a single person skiing in the image, and they are surrounded by an area with a lot of bushes.
===
Question:
What kind of skiing is the person doing? Cross-country or downhill?
===
Answer:
Based on what I can see in the photo, it is not possible to definitively determine whether the skier is doing cross-country or downhill skiing. However, given the presence of a trail sign and the fact that they are skiing in the foothills, it is more likely that the skier is engaged in cross-country skiing. Downhill skiing typically takes place on more steep slopes and involves ski lifts, whereas cross-country skiing often occurs on relatively flatter terrain with marked trails.
===
Question:
Can you explain cross-country and downhill skiing a bit more?
===
Answer:
Cross-country skiing and downhill skiing are two distinct types of skiing, each with its own techniques, equipment, and terrain preferences.

1. Cross-Country Skiing:
Also known as Nordic skiing, cross-country skiing involves traversing relatively flat terrain or gentle slopes using skis and poles. It is a popular recreational and competitive sport that focuses on endurance, balance, and aerobic fitness. Cross-country skiers use their own locomotion to move across snow-covered terrain, rather than relying on ski lifts or other forms of assistance.
There are two main techniques in cross-country skiing:
- Classic: This technique involves skiing in parallel tracks, using a forward-and-backward gliding motion while keeping the skis parallel to each other. The skier uses their poles to propel themselves forward.
- Skate skiing: This technique is similar to ice skating, where the skier pushes off one ski at an angle and glides on the other. It requires a smooth, groomed surface and is typically faster than the classic technique.
Cross-country ski equipment consists of lightweight skis, boots, bindings, and poles. The boots are more flexible than those used in downhill skiing, allowing for better ankle movement and control.

2. Downhill Skiing:
Also known as alpine skiing, downhill skiing involves descending slopes at high speeds using skis and poles for balance and control. This sport is more focused on speed, technique, and navigating challenging terrain, including steep slopes, moguls, and even jumps.
Downhill skiing can be further categorized into several disciplines, such as slalom, giant slalom, super-G, and downhill racing. Each discipline has its own set of rules, courses, and ski equipment.
Downhill ski equipment includes heavier and stiffer skis, boots, bindings, and poles than those used in cross-country skiing. The boots are more rigid to provide better support and control during high-speed descents and sharp turns.

In summary, cross-country skiing is an endurance-based sport that involves traveling across flat or gently sloping terrain, while downhill skiing is focused on speed and technique as skiers navigate steeper slopes and challenging terrain. Both sports require specialized equipment and techniques, but they offer different experiences and challenges to participants.

Table 16: One example used in in-context-learning to construct visual conversation data.

