# OpenReview forum: "Visual Instruction Tuning"
_NeurIPS.cc/2023/Conference — NeurIPS 2023 oral_

### Official Review · Reviewer_mhaS · 2023-06-26

**Soundness:** 2 fair
**Presentation:** 4 excellent
**Contribution:** 4 excellent
**Rating:** 6
**Confidence:** 5

**Summary:**

This paper introduces the first attempt to extend instruction-tuning paradigm to multimodal domain. This work has several major contributions: (a) the curation of the first vision-language instruction-following dataset by converting public image-text pairs into appropriate format using ChatGPT, resulting in over 100K+ multimodal instruction-following samples, (b) results indicating that a multimodal model (consisting of a CLIP visual encoder, a linear projection layer to convert visual tokens into language prompts, and a LLaMA language decoder) trained on this dataset can achieve robust multimodal chatting abilities. All assets used in their research including datasets and models are open-source.

**Strengths:**

There are two major novel contributions of this work: (1) it introduces one of the first large-scale instruction-following multimodal datasets by leveraging public image-text pairs, and (2) it releases all training code, pre-trained models, and evaluation benchmarks to the wider public. These assets (outlined in supplemental L64) are undeniably valuable to the multimodal research community.

**Weaknesses:**

I have several major concerns about (a) the evaluation benchmarks and metrics, (b) the lack of simple baselines such as captioning-based approaches, (c) missing implementation details such as the sampling procedure.

A: Issues about quantitative analysis for multi-modal chatting.

- The paper uses rather small evaluation sets (L217-235) to construct the LLaVA-Bench, including 30 randomly selected COCO images and 24 in-the-wild images. Why is this subset is much smaller than the pre-training dataset with 100K+ multimodal instruction-following samples? And how do you select the 24 in-the-wild images? I couldn't find evidence in the current draft to suggest that these 24 images are not cherry-picked.

- The evaluation is text-only and the authors use GPT4 to explicitly assign a score. While prior works such as Vicuna [1] also uses GPT4 to score their responses in a text-only fashion, it is unclear how robust is GPT4 for multimodal reasoning while doing text-only evaluation. For more robust quantitative analysis, I would encourage the authors to split the instruction-following datasets into train/val splits and also include the results of classic text-only scoring metrics. Small-scale human evaluation will also be beneficial.

B: Language prior of ScienceQA benchmark.

- I am shocked that a text-only (vision-blind) GPT4 can achieve as high as 82% accuracy on ScienceQA, suggesting that this particular VQA benchmark has severe language prior [2]. Even though prior works also adopt this benchmark in their evaluation, this makes it hard to interpret the progress achieve by LLaVA towards a truly “multimodal” instruction-following agent as this benchmark can be largely addressed by language prior information.
- Is it possible to report zero-shot LLaVA performance on ScienceQA?

C: Simple baselines such as dense captioning:

- Even though the model architecture of LLaVA looks elegant as it only uses a linear projection to connect CLIP’s visual tokens to soft language prompts, I believe an even simpler baseline is to train a dense captioning model (using the existing rich descriptions generated by prompting ChatGPT with caption+bounding box information). During inference time, the dense captioner can turn an image into a rich textual description, which can be sent to an instruction-following text-only LLM (Vicuna/GPT4).

D: Missing implementation details such as sampling.

- The sampling procedure (e.g., top-k/nucleus sampling/beam search) can have profound impact on the quality of generated texts. However, the current draft does not discuss how to perform sampling for LLaVA. Also, when using GPT4 for text-only evaluation, the exact hyperparameters used such as temperature should also be reported.

E: Generalization or bias?

- Fig. 5 in appendix suggests that LLaVA is able to generalize to unseen domains, i.e., correctly identifying that the person holding a doge coin is Elon Musk, while Elon Musk does not appear in LLaVA’s training dataset. However, it is unclear whether this is a result of generalization or language bias of LLMs. Perhaps your model tends to answer “Elon Musk” when asking about the name of the person, or perhaps it tends to answer “Elon Musk” when there is a doge coin in the image.


One minor typo:
L99: “and curate such a questions list” -> “to curate such a list of questions”


Finally, I have an doubt about "multimodal instruction-following" (this is not a weakness but open to discussion):

- Studies in NLP such as [3,9] have suggested that instruction-following is effective mostly because LLMs such as LLaMA are already capable foundation models, and therefore instruction-following can effectively align the model output with human interest. However, it is unclear whether multimodal foundation models such as CLIP (as used in LLaVA) is powerful enough. For example, a wide range of recent works and benchmarks [4,5,6,7,8] suggest that CLIP behaves like bag-of-words and do not have strong vision-language reasoning capabilities. As we do not yet have strong enough vision-language foundation models, it is unclear if the multimodal research community is ready to embrace the instruction-following paradigm.

[1] Vicuna. Vicuna: An open-source chatbot impressing gpt-4 with 90%* chatgpt quality.

[2] Making the V in VQA Matter: Elevating the Role of Image Understanding in Visual Question Answering. Goyal et al. 2016.

[3] LIMA: Less Is More for Alignment. Zhou et al. 2023.

[4] When and why vision-language models behave like bags-of-words, and what to do about it? Yuksekgonul et al. 2022.

[5] Winoground: Probing Vision and Language Models for Visio-Linguistic Compositionality. Thrush et al. 2022.

[6] CREPE: Can Vision-Language Foundation Models Reason Compositionally? Ma et al. 2022.

[7] Equivariant Similarity for Vision-Language Foundation Models. Wang et al. 2023.

[8] Visio-Linguistic Reasoning with Multimodal Generative Pre-Training Scores. Lin et al. 2023.

[9] The False Promise of Imitating Proprietary LLMs. Gudibande et al. 2023.


**Questions:**

I summarized my most concerned questions about this work:
- Why did you not sample train/val splits for evaluating LLaVA's multimodal chatting abilities? Are there specific concerns?
- Is it possible to extend LLaVA's evaluation to other VQA benchmarks (as reported by GPT4) such as VQA2.0 which has balanced language prior?
- Why is the architecture design of LLaVA more superior than a dense captioning model + instruction-following LLM, if both are trained on the same dataset?
- What is the sampling procedure of LLaVA?

Given that this paper presents a significant dataset contribution, I would be happy to revise my rating if the authors can address my above-mentioned weaknesses and questions.

Updated in Aug 17th: I have increased my rating based on the author's promise to revise the paper by including more discussion on more scientific evaluation metrics and benchmarks for MLLMs.

**Limitations:**

Yes, the authors discuss about limitations in supplemental material.

---

> ### Author Rebuttal · Authors · 2023-08-10
>
> **Q1. Why do we use a small LLaVA-Bench-COCO split with only 30 images?**
>
> Since we divide the question into three categories, we have 90 questions for COCO and 60 questions for In-the-Wild. The amount of the questions in our test set is similar to Vicuna-Bench [1], which has 80 questions. The reason we consider small evaluation sets is due to the cost using GPT-4 evaluation, the current size allows more accessibility for e.g., university labs to report results.
>
> **Q2. How do we select the 24 in-the-wild images?**
>
> Different from exsiting benchmark that evaluates a single specific capablity of multimodal models, we are seeking for a benchmark, where each evaluation sample can measure multiple integrated capabilities of a model. Such capabilities include recognition, OCR, knowledge, language generation, spatial awareness, counting, etc.  To design a benchmark covering a wide range of capabilities, while keeping the evaluation cost affordable, we find 24 samples that require multiple multimodal capabilities. Besides, by using the description-based annotations for each image, we are able to create and extend more questions to improve the capability coverage.
>
> **Q3. Split to train/val splits**
>
> LLaVA-Bench-COCO follows the train/val split in machine learning. LLaVA-Instruct-158K uses COCO train 2014 images and annotations, while it samples 30 validation images from COCO val 2014 (L217). Both follow the same data generation pipeline (Sec. 3). It serves as validation to study model alignment and capabilities with consistent visual inputs (L219-L220). A small size is chosen, similar to minival split practice, for quick evaluation / iteration during development.
>
> **Q4. Language-prior in ScienceQA and zero-shot performance**
>
> ScienceQA has three question modes: text, image, and no context (Table 7, TXT/IMG/NO).
>
> The default evaluation pipeline of ScienceQA includes a "random guess" mechanism, which helps text-only GPT-4 but not LLaVA. Without this, GPT-4's IMG accuracy drops to 59%, suggesting a more balanced language prior.
>
> As suggested by the reviewer, we provide two further experiments: (1) zero-shot performance of LLaVA and Vicuna; (2) Vicuna finetuned on ScienceQA.
>
> Since open-source models like Vicuna and LLaVA are still not good at following “format” instructions like "conclude your answer with `The answer is`", we use ChatGPT-3.5 to reformat the answer, following the recent practice in multimodal evaluation.
>
> We show that zero-shot LLaVA outperforms both GPT-4 (+7.5%) and Vicuna (+10.3%) on IMG modality. Besides, zero-shot LLaVA consistently outperforms zero-shot Vicuna, in all categories including text-only questions.
> We also show that the finetuned LLaVA outperforms the finetuned Vicuna in almost all categories, with an average performance gain of 5.2%.
>
> | ZeroShot| NAT | SOC| LAN| TXT| IMG| NO| G1-6 | G7-12 | AVG|
> |--|--|--|--|--|--|--|--|--|--|
> | GPT-4 | 77.44 | 64.23 | 86.45 | 74.73 | 59.05 | 90.38 | 78.49 | 74.36 | 77.01 |
> | Vicuna | 66.83 | 60.63 | 69.00 | 65.69 | 56.27 | 71.71 | 68.94 | 60.98 | 66.09 |
> | LLaVA  | 71.27 | 74.24 | 70.91 | 70.43 | 66.53 | 72.89 | 75.4 | 65.33 | 71.8 |
>
> | Finetuned | NAT | SOC| LAN| TXT| IMG| NO| G1-6 | G7-12 | AVG|
> |--|--|--|--|--|--|--|--|--|--|
> | Vicuna | 86.9 | 79.3 | 88.55 | 85.58| 76.85 | 91.43 | 85.68 | 85.83 | 85.73 |
> | LLaVA | 90.36 | 95.95 | 88 | 89.49| 88 | 90.66 | 90.93 | 90.9 | 90.92 |
>
> We'll include these results and discussion in the revision.
>
> **Q5. Dense captioning model + LLM vs end-to-end**
>
> There are two conceptual advantages.
>
> - Completeness of image representations. Dense captioning may not be able to capture all details of an image that the user instruction is concerned with. In contrast, an end-to-end multimodal model can be instruction-aware and only focus on the relevant visual contents via attention mechanism. See Rebuttal Fig. 2 for a qualitative example.
> - Single model. End-to-end models save computational resources and reduce complexity in model serving and request handling, which is beyond the scope of this paper.
>
> **Q6. Sampling Details**
> We set LLaVA's temperature to 0.2, and tuning other hyper-parameters does not further improve the quality. Beam search improves LLaVA-13B output quality (67.3->69.8), while it is not trivially compatible with real-time UI like ChatGPT. We report all numbers without the beam search to make the evaluation consistent with the user interface. For GPT queries, we follow Alpaca to set the temperature and top_p to 1.0. We will include this in revision.
>
> **Q7. Elon Meme: Generalization or bias**
> This is a great question that is worth further studying. We designed two sets of images in Rebuttal Fig. 1 to verify. We find this study intriguing and it further supports it being a form of generalization of our model.
>
> **Q8. Is CLIP powerful enough for multimodal instruction-following?**
>
> Thanks for bringing up this interesting topic.
>
> We agree with that instruction tuning plays a more important role in guiding capable foundation models to follow human intents, rather than adding new knowledge. In multimodal settings, we leverage the existing capabilities of two models: LLM for language knowledge and CLIP for image-text alignment, rather than making each individual of them stronger. LLaVA tuning largely aligns the capable foundation LLM to understand image-related human intent. Interestingly, new emerging properties occur when combining the two existing capabilities like OCR in the wild.
>
> Grid features or the pooled features on [CLS] token of CLIP on their own may lose spatial information (behave like bag-of-words). However, in LLaVA’s design, we feed the raw image patch features into the LLM, and the positional embeddings of LLM are incorporated into the visual token representations, implicitly maintaining the spatial information of the visual inputs.
>
> [1] Chiang, et al. "Vicuna: An open-source chatbot impressing gpt-4 with 90%* chatgpt quality.".

---

> > ### Comment · Reviewer_mhaS · 2023-08-10
> > **Follow-up questions**
> >
> > Thanks for the detailed response. Some of my questions in the original review were not answered and I would still appreciate a discussion on these topics:
> >
> > **Q1. "While prior works such as Vicuna [1] also uses GPT4 to score their responses in a text-only fashion, it is unclear how robust GPT4 is for multimodal reasoning while doing text-only evaluation."**
> >
> > After reading the response, I am even more concerned about GPT-4 text-only evaluation:
> > - It costs money to perform evaluation, which limits the custom test-set in this paper to 30 COCO and 24 in-the-wild images.
> > - GPT-4 performance is not stable across time.
> > - As mentioned by authors in response to Q5, the textual annotation of an image may not capture all the details in an image.
> > - While prior works such as Vicuna also use GPT4 to output numerical scores, it is hard to measure how accurate the numerical score is.
> >
> > **Q2. "Are the 24 in-the-wild images cherry-picked?"**
> >
> > By cherry-picking I actually meant **selecting test samples based on the performance of LLaVA**. While I would like to believe that the authors did not do this, this evaluation set just seems too small for robust and scientific benchmarking. Although it covers a diverse range of skills, I would like the authors to provide more discussion in the revised paper about **what is the scientific way to benchmark such multimodal instruction-following models.** For example, should one be using larger and standardized benchmarks such as VQA2.0 and GQA (or some more recent multimodal benchmarks)? Are there better and more reproducible evaluation metrics than GPT-4's raw numerical scores? Because LLaVA is one of the first works on this trending topic, I believe it is important to answer these questions such that the community can move in the right direction.
> >
> > After reading the response to Q4, I would like the authors to clarify a few things about ScienceQA experiments:
> >
> > **Q3. What is the "random-guess" mode, and why does it help with text-only GPT4? I couldn't find the term "random guess" in the original ScienceQA paper or their github repo, so I would like a more detailed explanation here.**
> >
> > **Q4. How does the text-only GPT4 and zero-shot/fine-tuned Vicuna baseline utilize the image context? Do they use the image caption from the ScienceQA dataset?"**

---

> > > ### Author Response · Authors · 2023-08-12
> > > **Discussion: the scientific way to benchmark multimodal instruction-following models**
> > >
> > > We thank the reviewer for the insightful comments, and we are happy to discuss the evaluation aspect for multimodal models.
> > >
> > > For existing benchmarks, they usually focus on a single aspect. However, what is unique to the recent large multimodal models (LMM), like multimodal GPT-4, is that they can perform visual tasks in the wild that require integrated capabilities. For example, in Supplementary Table I, to correctly complete the user’s request, it requires the model to have: (1) OCR capability to understand the captions; (2) visual recognition capability to understand that it is a pan of nuggets that looks like a world map; (3) reasoning capability to combine the information together and answer why this can be interesting.
> > >
> > > **Having a model that excels in each single aspect, does not necessarily guarantee a capability to combine and reason about them in a single answer**. For example, BLIP-2 is one of the best models that ranks top across the board of academic benchmarks, while lacking the complex reasoning capabilities. Further evidence can be found in Table 5, where BLIP-2 is capable at answering short-form “conversational” questions and fails to tackle more complex tasks. Some recent evaluation benchmarks on LMMs also revealed similar results.
> > >
> > > At the time of submission, such a benchmark was lacking. This motivates us to construct such a benchmark that requires the model to leverage different capabilities to correctly complete a user’s task. We tried to utilize the resources we have at the time of submission, to construct such a benchmark, with an aim to create scientific and controlled settings. **We do not cherry-pick test samples based on the performance of LLaVA. We do not tune any model/data design choices, based on any of the results we obtain on LLaVA-Bench-In-the-Wild, and we use that solely for benchmark purposes**.
> > >
> > > However, we fully agree with the reviewer that to have a more comprehensive and complete understanding of the model’s capability, we would need a benchmark at a larger scale, which we were unable to achieve due to the cost limit.
> > >
> > > Some recent works show a clever way to address this. Instead of directly evaluating with GPT-4, one can use ChatGPT to extract the answers, or key aspects that are required to solve the problem. This can allow the model to still answer with natural sentences, while enabling evaluation at scale. We are happy to see such progress in this field. Of course, despite being cheaper, ChatGPT still incurs a cost. During the rebuttal, we find the largest open-source model, LLaMA-2-Chat-70B, elicits impressive capabilities in following complex instructions like ones we used to query GPT-4 to create LLaVA-Instruct-158K (see response to Q1 of MLCz). It can also be a cost-free alternative.
> > >
> > > Meanwhile, we believe that it is also important to leverage academic benchmarks, like VQAv2, GQA, etc. The current LLaVA is only trained on natural instruction and responses, making it challenging to evaluate on those standard benchmarks that have ground truth with a single or a few words. Given the large size of these datasets, it is an open research problem to design efficient and effective metrics. We find the recent VisualGPTScore [1] can be considered as an inspiring way to construct a cost-efficient metric. For example, one can evaluate $P(text|image)$ after the model outputs the long-form answer.
> > >
> > > We thank the reviewer for bringing up this important topic for discussion, and we are happy to discuss and clarify any further questions or doubts. We will include these discussions in our revision, and we believe our draft will be stronger by including the insightful suggestions from the reviewers.
> > >
> > > [1] Visio-Linguistic Reasoning with Multimodal Generative Pre-Training Scores. Lin et al. 2023.

---

> > > > ### Author Response · Authors · 2023-08-12
> > > > **Response to follow-up questions**
> > > >
> > > > **Q1. "While prior works such as Vicuna [1] also uses GPT4 to score their responses in a text-only fashion, it is unclear how robust GPT4 is for multimodal reasoning while doing text-only evaluation."**
> > > >
> > > > Below, we further clarify some aspects of GPT-4 evaluation, and the measures that we have taken to make sure that the experiments are conducted under a controlled setting.
> > > >
> > > > > It costs money to perform evaluation, which limits the custom test-set in this paper to 30 COCO and 24 in-the-wild images.
> > > >
> > > > We agree. We have discussed some alternatives above.
> > > >
> > > > > GPT-4 performance is not stable across time
> > > >
> > > > We find the comment relevant to the release of an iteration of GPT-4 on June 13. We do want to note that the gpt-4-0613 is released after the NeurIPS submission deadline, and we have taken two measures to make sure the experiments are conducted in a controlled setting.
> > > >
> > > > We fixed the GPT-4 endpoint version to gpt-4-0314, after the release of gpt-4-0613. The same endpoint outputs consistent evaluations as verified in Table 5. This ensures that the evaluation is consistent before the retirement of the gpt-4-0314 endpoint (no earlier than July 5, 2024). We found that the relative rankings of different methods remain consistent as long as the same API is used (across gpt-3.5-turbo-0301, gpt-4-0314, gpt-4-0613, respectively).  We suggest that future researchers should fix the same API and evaluation pipeline throughout their experiments for all the methods, and report API endpoints in the paper.
> > > >
> > > > > As mentioned by authors in response to Q5, the textual annotation of an image may not capture all the details in an image.
> > > >
> > > > We would like to clarify that when we construct the instructions in the LLaVA-Bench, we make sure that the instruction is answerable by referring to the provided image context.
> > > >
> > > > In the context of Q5, we want to emphasize that, when the model is generating captions, without the access to the user instruction, it may not capture all details in the image that the user cares about. We believe the instruction-aware property can be of great value to future research.
> > > >
> > > > > While prior works such as Vicuna also use GPT4 to output numerical scores, it is hard to measure how accurate the numerical score is.
> > > >
> > > > MTBench [1] evaluates the agreement between GPT-4 scores and the human evaluation, and shows that there is a strong agreement between them. A recent study to evaluate LMM shows that the agreement between GPT-4 score and human evaluation is quite consistent (4.2% absolute discrepancy).
> > > >
> > > > **Q2.**
> > > > Answered with the discussion above.
> > > >
> > > > **Q3. Clarification of "random-guess" mode.**
> > > >
> > > > We apologize for the confusion. In the official implementation (`models/run_gpt3.py#L70`), when the model does not output a valid answer, it randomly picks an answer. This improves the GPT-4’s accuracy when it refuses to answer. For fine-tuned LLaVA models, it follows the predefined format, so adding this does not further improve the model’s performance.
> > > > When we evaluate zero-shot LLaVA and Vicuna, such cases also happen. We also treat them as incorrect and do not perform random choice.
> > > >
> > > > **Q4. How does the text-only GPT4 and zero-shot/fine-tuned Vicuna baseline utilize the image context? Do they use the image caption from the ScienceQA dataset?"**
> > > >
> > > > We do not add image captions from the ScienceQA dataset.
> > > > We made this choice as we find that adding the image caption harms the Vicuna’s zero-shot performance (IMG 56.27 -> 52.95).
> > > > We hypothesize that it can be due to the fact that the image caption does not always provide accurate context (similar to observations in Fig. 17 and relevant discussions in ScienceQA paper). This can confuse the model and thus make incorrect predictions, after such irrelevant context is incorporated.
> > > >
> > > > [1] Zheng, et al. "Judging LLM-as-a-judge with MT-Bench and Chatbot Arena."

---

> > > > > ### Comment · Reviewer_mhaS · 2023-08-17
> > > > > **Thank you for the detailed discussion**
> > > > >
> > > > > I genuinely thank the authors for their discussion on more scientific and cost-efficient evaluation metrics for MLLMs. I believe such a revision of the paper will make the submission much stronger, and thus I have increased my rating. I also appreciate authors' detailed response on all my other questions.
> > > > >
> > > > > The above response seems to be missing a few references. For example:
> > > > >
> > > > > 1 - *"Some recent works show a clever way to address this. Instead of directly evaluating with GPT-4, one can use ChatGPT to extract the answers, or key aspects that are required to solve the problem."* -- What are the recent works?
> > > > >
> > > > > 2 - *"A recent study to evaluate LMM shows that the agreement between GPT-4 score and human evaluation is quite consistent (4.2% absolute discrepancy)."* -- What is the recent study?

---

> > > > > > ### Author Response · Authors · 2023-08-18
> > > > > >
> > > > > > We sincerely appreciate the reviewer's insightful comments and the increased rating.
> > > > > >
> > > > > > Regarding the missing citations, we would like to clarify that the paper is cited by those publications. We have consulted with the ACs, and they advised us not to include citations of these papers to maintain the anonymity of the review process. We will provide the exact references to the ACs and PCs. Once the reviewing process is concluded, we will post the citations here via an "Official Comment" for your reference and for the benefit of the community.

---

### Official Review · Reviewer_A2dU · 2023-07-06

**Soundness:** 3 good
**Presentation:** 3 good
**Contribution:** 4 excellent
**Rating:** 8
**Confidence:** 4

**Summary:**

This presents a multi-modal instruction following model and evaluates it. The model is trained by using a frozen vision encoder whose features are used an input to a LLM, which is fine-tuned. It is trained first on simple captioning tasks using a large amount of data, and then on a multi-modal instruction following dataset built by having a LLM generate tasks using the captions and bounding boxes of images.

**Strengths:**

- Presents an impressive model that seems to have advanced instruction following abilities for images and text.
- Present a detailed set of comparison and evaluation using LLaVA-Bench, and the ScienceQA experiments. I also appreciated the re-use of a qualitative example from the GPT-4 paper.
- Some interesting insights, such as using the second to last layer and getting a sense of the benefit of each kind of instruction-tuning task.

**Weaknesses:**

- I think it would be valuable to see if this same pipeline could work with an open source LLM, using OpenAI's closed LLMs are not ideal for scientific understanding and reproducibility. I in particular wonder if other LLMs can use the bounding boxes annotation of the images effectively, which I imagine in much harder than the using the captions.
- Seeing an evaluation on standard vision/language benchmarks would have been interesting, such as zero-shot VQA.


**Questions:**

Would the data generation pipeline benefit from even more detail image descriptions? There are other annotation like region captions or visual narratives that could have also been used.

I am also curious if the model can handle very low-level detailed questions, like "What color is the rope the man is using?" for Table 3.

278: year -> layer

**Limitations:**

The authors have a detailed section in the appendix.

---

> ### Author Rebuttal · Authors · 2023-08-10
>
> **Q1. Does the proposed pipeline work with an open-source LLM?**
>
> In our preliminary study, we find that the capability of the teacher is crucial to the quality of the generated instruction-following data (L128-L130). Until the submission deadline, the largest Vicuna model was 13B. Just as the reviewer’s concern, its complex reasoning and spatial reasoning capability is still limited and is behind proprietary models including ChatGPT and GPT-4. However, the recently released LLaMA-2-70B-Chat appears to have narrowed the gap. Due to the large size of the model, it requires a huge amount of VRAM and has a slow inference speed. We conducted a preliminary study on around 200 samples for each category.
>
> Specifically, we generate 200 samples for each category (conversation, detailed description, complex reasoning), using LLaMA-2-70B-Chat, ChatGPT, and GPT-4.
>
> After generating the response, we find that unlike previous open-source models, LLaMA-2-70B-Chat can start to follow complex instructions like creating multimodal instructions.
>
> However, it still fails in the conversation category, as we find that the LLaMA-2-70B-Chat is not correctly following the conversation format. This may be potentially fixed with more sophisticated prompt tuning. However, due to the limited rebuttal period, we do not evaluate on the conversation category. This is also one of the main limitations we find of LLaMA-2-70B-Chat.
>
> We then quantitatively evaluate the generated instructions using GPT-4 as the judge: (1) the correctness of the answers generated, and (2) the complexity of the instructions generated for complex reasoning questions.
>
> |          | Correctness | Complexity |
> |----------|-------------|------------|
> | LLaMA-2-70B-Chat | 8.7         | 7.4        |
> | ChatGPT  | 9.5         | 9.2        |
>
> These initial results are promising, and suggest that our pipeline can be potentially applied to open-source models as their capabilities are improved. We look for more comprehensive studies and deeper explorations for future research.
>
> **Q2. Zero-shot VQA**
>
> We evaluate LLaVA-13B on VQA-v2 and OK-VQA.
>
> Note that for OKVQA, we use a slightly relaxed evaluation protocol for evaluation. Since LLaVA typically outputs a short sentence rather than one or two words, if the generated sentence by LLaVA contains the ground truth answer, we consider it a correct prediction.
> We find that although LLaVA lags behinds on VQA-v2 (a task where answers can be directly derived from the image), LLaVA performs surprisingly well on OK-VQA (a task where answers require strong knowledge and reasoning) — LLaVA outperforms Flamingo-80B on zero-shot OK-VQA. This is because (1) LLaVA’s advantage lies in its capable LLM; (2) all of the instructions in LLaVA are prompting the model to output a complete sentence, and it thus struggles a bit on those standard benchmarks, which requires answers of one or two words.
>
> We believe the latter issue can be alleviated by incorporating short-form answers into the instruction tuning data or improving the multimodal in-context learning capabilities, and we leave them to future work.
>
> | Models       | VQAv2 | OKVQA |
> |--------------|-------|-------|
> | PICa (in-context few-shot prompting GPT3-175B) [1]  | --  |48.0  |
> | Flamingo-80B [2] | 56.3  | 50.6  |
> | LLaVA-13B    | 44.2  | 55.0  |
>
> [1] PICa: An Empirical Study of GPT-3 for Few-Shot Knowledge-Based VQA
>
> [2] Flamingo: a visual language model for few-shot learning
>
> **Q3. Will the data generation pipeline benefit from even more detailed image descriptions?**
>
> This is a great suggestion. As we show in Table 4, adding detailed image description to conversation data can improve the model’s capability. We believe that more detailed annotations can further improve the detailedness and the quality of the generated instruction dataset, ultimately resulting in an improved model. Annotations, such as region captions, visual narratives, or scene graphs, are definitely valuable for future research.
>
> **Q4. Can LLaVA handle questions about extremely low-level details?**
>
> Currently, LLaVA is not able to handle low-level details such as the color of the rope, as the rope is currently very thin and is covering only a few pixels after center cropped and resized to 224x224. However, given that LLaVA is capable of correctly identifying the color of those slightly larger regions (e.g. the clothes that the man is ironing), we believe that scaling up the image resolution, and/or incorporating more detailed descriptions, as mentioned by the reviewer in the comment above, could unlock LLaVA’s low-level detail recognition/reasoning capabilities.
>
> > 278: year -> layer
>
> Thanks for pointing out the typo. We’ll fix it in the revision.

---

> > ### Author Response · Authors · 2023-08-20
> >
> > Dear reviewer, we would like to thank you for your insightful feedback. We hope that your questions are addressed with our rebuttal. Please let us know if there are any further questions that need clarification.

---

> > ### Comment · Reviewer_A2dU · 2023-08-21
> >
> > Thank you for answering my questions, I think discussion about the weaknesses of these systems is also very valuable. I continue to be very positive about this paper.

---

### Official Review · Reviewer_MLCz · 2023-07-07

**Soundness:** 4 excellent
**Presentation:** 4 excellent
**Contribution:** 4 excellent
**Rating:** 8
**Confidence:** 5

**Summary:**

This paper studies instruction tuning in the multimodal domain. Instruction tuning has recently drawn a lot of attractions in the large language model (LLM) field, and hence it is interesting and important to study similar capabilities in multimodal models. This paper is a pioneer work in this direction. It constructs the first instruction tuning dataset LLaVA and conducts comprehensive analysis. This is an important step towards general-purpose multimodal language model (MMLM) that can interact with humans using natural languages.

**Strengths:**

1. To the best of my knowledge, this is the first work of multimodal instruction tuning. This is an important direction and hence the impact of this paper is huge.
2. The dataset constructed in this paper is both useful and inspiring. It constructed an impactful first step for future studies in this field.
3. The paper is well written and contains a lot of detailed studies.

**Weaknesses:**

I don't see any major weakness of the paper. There are some minor ones that can be improved but I understand they may go beyond the scope of this paper. For example, it will be more comprehensive to conduct ablations on the models used in this paper other than Vicuna etc.

**Questions:**

What did you use 2e-3 lr for pretraining? This seems to be a very large value given the 128 batch size.

---

> ### Author Rebuttal · Authors · 2023-08-10
>
> **Q1. Ablations on other LLMs other than Vicuna.**
>
> Until the paper submission deadline, Vicuna is the most adopted open-source instruction-tuned LLM. There are other great instruction-tuned LLMs coming out after that, including MPT, LLaMA-2-Chat, etc.
>
> We present initial studies using these other LLMs on LLaVA-Bench-In-the-Wild below. We also include the performance of the base LLM measured by MMLU and MTBench from the Vicuna leaderboard.
>
> | BaseLLM | MMLU | MTBench | LLaVA-Bench-In-the-Wild |
> |------------|------|---------|----------|
> | MPT-7B | 32      | 5.42 | 53.6 |
> | Vicuna-7B | 49.8    | 6.17 | 63.3 |
> | LLaMA-2-7B-Chat      | 45.8    | 6.27 | 63.2 |
>
> This initial study exhibits a correlation of the capability of the base LLM and the performance of the resulting multimodal model performance. It will be interesting to dig further into the relationship between the base LLMs and multimodal models. It is worth mentioning that MPT-7B and Vicuna-7B is instruction-tuned with supervised finetuning, while LLaMA-2-7B-Chat is additionally finetuned with RLHF. It will also be interesting to see the influence of RLHF in terms of the multimodal capabilities.
>
> We believe this initial study will be valuable to the research community to better understand the mechanism and capability of the multimodal models. We will release these results and corresponding model checkpoints to the public.
>
> **Q2. Why do we use a 2e-3 learning rate for pretraining?**
>
> Since we pretrain our model on a small 595K dataset, there are only around 4.5K iterations during training. Given the few steps the model is optimized for, we empirically find a larger 2e-3 learning rate is slightly better for fast convergence than lower learning rates like 2e-4.

---

> > ### Author Response · Authors · 2023-08-20
> >
> > Dear reviewer, we would like to thank you for your insightful feedback. We hope that your questions are addressed with our rebuttal. Please let us know if there are any further questions that need clarification.

---

### Official Review · Reviewer_RuBm · 2023-07-08

**Soundness:** 4 excellent
**Presentation:** 3 good
**Contribution:** 3 good
**Rating:** 5
**Confidence:** 4

**Summary:**

This paper introduced LLaVA, an effective visual instruction tuning method to turn Large Language Models (LLMs) into multi-modal LLMs. LLaVA is first pre-trained on image-text pairs to connect a visual encoder (CLIP) and a LLM (Vicuna). Then the authors utilize GPT4 to generate ~150K visual instruction data for training visual instruction-following models. LLaVA is evaluated on two diverse and challenging benchmarks, as well as a science question answering dataset. Overall, LLaVA is a very early attempt for enhancing LLMs with multi-modal capacity. I believe it will inspire a lot to the research community.

**Strengths:**

- Expand instruction tuning to the vision domain. Visual instruction tuning is a new research problem for vision-language models. It endows vision-language models with powerful comprehension and reasoning capabilities.
- A new pipeline for visual instruction data generation. LLaVA takes image captioning or object detection results as GPT4’s input for visual instruction generation. This is an effective way to quickly generate a large amount of visual instruction data.
- A strong visual instruction model LLaVA with available pretrained models and demos.
- Multi-modal instruction-following benchmark.

**Weaknesses:**

- Data quality. The proposed visual instruction data is automatically generated by GPT4. But there seems to be a lack of validation of the data quality. For example, MiniGPT4 [1] will manually verify the correctness of each image description. High quality instruction data is also proved to be important for pure LLMs in LIMA [2].
- Is 595K image-text data enough for vision-language alignment? Blip2 uses >100M image-text pairs for vision-language alignment, while LLaVA only uses 595K data from CC3M. Have the authors tried to use more pre-training data, and will the model be further improved? Besides, I noticed that LLaVA inputs 256 CLIP visual tokens into the LLM, which is much large than Blip2 and MiniGPT4 (~30 tokens). Such a design will make the training much slower. So, do we really need 256 tokens?
- The LLM in LLaVA is fully fine-tuned in the second stage. Will this lead to degradation of LLM's ability? Are there verification results on traditional LM tasks?
[1] Enhancing Vision-Language Understanding with Advanced Large Language Models
[2] LIMA: Less Is More for Alignment.

**Questions:**

See Weakness

**Limitations:**

Yes. The authors have addressed the limitations and potential negative societal impact.

---

> ### Author Rebuttal · Authors · 2023-08-10
>
> **Q1. Data Quality**
>
> We agree that high quality instruction data is critical and have taken measures to ensure the data quality.
>
> First, we create image descriptions directly from the well-established manually-annotated MSCOCO dataset, which contains bounding box and caption annotations (L103-L110). This ensures the quality of the visual groundness of our textural context input to GPT-4.
>
> Second, we perform text-based filtering to remove invalid responses: (1) incomplete responses; (2) GPT refuses to provide the answer; (3) contain words that make the answer not sound like it is looking at the images (e.g. according to the captions). We’ll include the detailed list of keywords in the revised appendix.
>
> Third, we iterate and validate our prompts on a subset of around 1000 samples, to validate the visual groundness of the generated outputs using GPT-4, and find that GPT-4 consistently provides higher quality instruction-following data (L128-L130).
>
> Finally, we ablate using LLaVA-Bench-COCO on the combination of different types of generated instruction-following data (Table 4, L217-L227).
>
> Note that MiniGPT-4 is a concurrent work to LLaVA, while we are more than happy to discuss.
>
> > *From the MiniGPT-4 paper: ...we check if each generated image description **follows our desired format**, and also manually refine the generated captions by **eliminating redundant words or sentences** ....*
>
> According to the MiniGPT-4 paper, the manual check is only performed to correct **textual format** errors, without mentioning checking the visual groundness of the generated responses.
>
> Furthermore, the source image description of MiniGPT-4 is generated by the first-stage MiniGPT-4. Besides the textual errors that are mentioned in the paper, the correctness and the visual groundness of the generated descriptions of MiniGPT-4 is unclear.
>
> We believe that our data is more large-scale, diverse, content-rich, and the quality is more controlled.
>
> **Q2. Is 595K image-text pairs enough for vision-language alignment?**
>
> Since CC3M only has around 2M images available to download from the Internet, we choose the BLIP-captioned LAION-CC-SBU dataset (which is the training dataset of BLIP2). Due to the limitation of both resources and time during the rebuttal period, we have tried two subsets: 600K samples and 6M samples. We ablate this with Vicuna-13B using the same schedule as described in the paper, and evaluate on LLaVA-Bench-In-the-Wild. As shown in the paper, when scaling up the pretraining dataset from 600K to 6M, the overall performance on LLaVA-Bench-In-the-Wild does not vary too much (66.8 vs 66.5).
>
> | Pretrain Samples | Conversation | Detail | Complex | All |
> |--|--|--|--|--|
> | 600K | 56.7±3.9 | 54.2±3.1 | 80.2±1.5 | 66.8±0.8 |
> | 6M | 55.1±3.9 | 56.3±2.6 | 79.6±2.9 | 66.5±0.5 |
>
> We believe the fast alignment of LLaVA can mainly be attributed to two reasons. First, our vision encoder, CLIP, was pretrained with image-text contrastive loss, and its visual feature is thus already aligned to a text space. It is sufficient to re-align this to a different text space using a linear layer. Second, it is much easier and requires fewer samples to optimize the linear layer (5.2M parameters) than the QFormer from BLIP2, which contains 1.1B parameters, orders of magnitude more than LLaVA’s alignment-stage trainable parameters. We thank the reviewer for bringing up this topic, and will include this discussion in the revision.
>
> **Q3. LLaVA inputs 256 CLIP visual tokens into the LLM, which is much larger than Blip2 and MiniGPT4 (~30 tokens). Such a design will make the training much slower. So, do we really need 256 tokens?**
>
> We are happy to compare with the concurrent work, MiniGPT-4.
>
> *First, will this make the training slower?*
>
> Since LLaVA uses 256 tokens, it is around 4-5x slower than MiniGPT-4 per training iteration. However, since we only need 600K samples to converge, the total pretraining cost of LLaVA is 4 hours on 8x A100s (Supp. L62). MiniGPT-4 pretrains with ~6M image-text pairs and requires training approximately 10 hours on 4x A100s (which roughly equates to 5 hours on 8x A100s). When considering the total training time, LLaVA is slightly faster.
>
> *Second, do we really need 256 tokens?*
>
> This is an interesting research question open to discussion. Compressing 256 tokens to 32 tokens is a process of information compression. We find that this is detrimental in terms of OCR capability, which is an interesting emergent capability of LLaVA. For example, on a suite of 27 text recognition related academic datasets,  LLaVA consistently outperforms MiniGPT4 on 23 out of 27 datasets, despite LLaVA being trained with an order of magnitude smaller image-text training data.
> We also qualitatively show in Fig. 2 of Rebuttal Supplementary, that such compression process may discard information that the user is curious about: LLaVA recognizes the website that the image comes from by reading the text from the watermark, while MiniGPT-4 fails.
>
> Furthermore, having finer patch-level features can allow the model to perform region-level reasoning easier, as the region-level information is better preserved and readily extractable for downstream models.
>
> **Q4. Does full-model finetuning lead to the degradation of LLM’s ability?**
>
> We show that LLaVA and Vicuna are comparable on MTBench[2], and LLaVA is only slightly worse (-0.8%) on MMLU[1].
>
> | | MTBench | MMLU |
> |---|---|--|
> | Vicuna-13B | 6.57  | 55.8 |
> | LLaVA-13B | 6.63  | 55.0 |
>
> We find this result encouraging, and this can be partially attributed to the inclusion of complex reasoning questions, and long-form answers in LLaVA-Instruct-158K, which helps maintain the language capabilities of LLaVA. We also show that on ScienceQA, it even slightly outperforms Vicuna, on text-only categories. See detail in R4 to reviewer mhaS.
>
> [1] Hendrycks, et al. "Measuring massive multitask language understanding.".
> [2] Zheng, et al. "Judging LLM-as-a-judge with MT-Bench and Chatbot Arena.".

---

> > ### Author Response · Authors · 2023-08-20
> >
> > Dear reviewer, we would like to thank you for your insightful feedback. We hope that your concerns are addressed with our rebuttal. Please let us know if there are any further questions that need clarification.

---

### Author Rebuttal · Authors · 2023-08-10

We sincerely thank all the reviewers for their time and their thoughtful comments and questions. We are encouraged that the reviewers find that:

- Our work is a pioneer in the multimodal instruction tuning field (RuBm, MLCz, A2dU, mhaS). It will inspire a lot to the research community (RuBm) and have a huge impact on this field (MLCz).
- We have made significant contributions, including
    - an inspiring pipeline for multimodal instruction data generation (RuBm,MLCz)
    - one of the first large-scale vision-language instruction-following datasets (mhaS) and a multimodal instruction-following benchmark (RuBm)
    - LLaVA, a strong visual instruction model (RuBm) with elegant designs (mhaS) and impressive instruction following capabilities for images and text (A2dU).
    - fully open-sourced assets that are undeniably valuable to the multimodal research community (mhaS).
- The paper is well written (MLCz), contains some interesting insights (A2dU), and has a detailed set of comparisons and evaluations (MLCz,A2dU).

We attempted our best to address the questions as time allowed. We believe the comments & revisions have made the paper stronger and thank all the reviewers for their help. Please find individual responses to your questions below.

---

### Decision · Program_Chairs · 2023-09-21

**Decision:**

Accept (oral)

**Comment:**

This paper introduces LLaVa, an multimodal large language model that visual instructionally fine-tuned.  LLaVA is first pre-trained on image-text pairwise data to align the image encoder and a pre-trained LLM. In the second step, the authors fine-tune the model using ~150k visual instruction data automatically generated by GPT4. All code and data are all open sourced to the research community.

The reviewers comment out this paper is one of the first work of multimodal instruction tuning, pioneering on the direction of tuning a pre-trained LLM for a better cross-modal alignment and understanding.  One major contribution is the 150k multimodal instruction tuning data. The idea of using GPT4 to generate instruction following data for multimodal is interesting and inspiring. The results are also strong and encouraging, indicating the approach of instruction tuning a LLM to accept multimodal information is a promising direction. Although one weakness, as reviewer A2dU pointed out,  is that the current dataset construction process is largely depend on the closed-source GPT4, and it still remains unclear how this can be fully reproduced using open sourced LLMs. The paper still deserves its merit to pioneering the approach and excellent experience to demonstrate the path is valid.

The AC carefully reads the submission, comments, and authors’ responses. The paper is also well written and structured well. Authors have been very engaged and answer all reviewers’ question well with reasonable thoughts and numbers. The AC generally agrees the reviewer’s comment and appreciates the fruitful discussions. The paper is an important step towards general purpose multimodal language model. The open sourced data and code will also benefit the community and motivate the research in this direction.

Therefore, the AC recommends to accept this paper.